# The Use of Additive Manufacturing Techniques in the Development of Polymeric Molds: A Review

**DOI:** 10.3390/polym16081055

**Published:** 2024-04-11

**Authors:** George Pelin, Maria Sonmez, Cristina-Elisabeta Pelin

**Affiliations:** 1INCAS—National Institute for Aerospace Research “Elie Carafoli”, Bd. Iuliu Maniu 220, 061126 Bucharest, Romania; pelin.george@incas.ro; 2INCDTP-ICPI—National Research and Development Institute for Textile and Leather—Division Leather and Footwear Research Institute, Ion Minulescu St. 93, 031215 Bucharest, Romania; maria.sonmez@icpi.ro

**Keywords:** additive manufacturing, 3D-printed molds, polymeric materials

## Abstract

The continuous growth of additive manufacturing in worldwide industrial and research fields is driven by its main feature which allows the customization of items according to the customers’ requirements and limitations. There is an expanding competitiveness in the product development sector as well as applicative research that serves special-use domains. Besides the direct use of additive manufacturing in the production of final products, 3D printing is a viable solution that can help manufacturers and researchers produce their support tooling devices (such as molds and dies) more efficiently, in terms of design complexity and flexibility, timeframe, costs, and material consumption reduction as well as functionality and quality enhancements. The compatibility of the features of 3D printing of molds with the requirements of low-volume production and individual-use customized items development makes this class of techniques extremely attractive to a multitude of areas. This review paper presents a synthesis of the use of 3D-printed polymeric molds in the main applications where molds exhibit a major role, from industrially oriented ones (injection, casting, thermoforming, vacuum forming, composite fabrication) to research or single-use oriented ones (tissue engineering, biomedicine, soft lithography), with an emphasis on the benefits of using 3D-printed polymeric molds, compared to traditional tooling.

## 1. Additive Manufacturing Introduction

Additive manufacturing (AM), commercially as known 3D printing, rapid prototyping, solid freeform fabrication, rapid manufacturing, desktop manufacturing, direct digital manufacturing, layered manufacturing, generative manufacturing, tool-less model making, etc., originates in the principles of topography and photo sculpture that uses a layered method to create 3D-shaped objects [1]. Additive manufacturing research studies were first conducted in the 1960s; techniques based on it were first commercialized around the 1980s by 3D Systems company [2], and since then, it is under constant growth and evolution. The layer-by-layer principle creates the most powerful advantage of AM, which is the ability to create almost any possible shape, while decreasing the time of product development, making it a solution to build complex and exotic structures that are difficult to achieve with conventional manufacturing strategies [2,3]. Besides the complexity of geometries, AM is promising due to the rapid production time, low to zero waste, and reduced labor costs, with high precision and accuracy [4,5].

Although the most popular term “3D printing” is often used to refer to additive manufacturing, in fact there are several individual layers processing manufacturing, depending on the materials, machining, and design used. Besides the technological process that develops with the aid of computer-assisted automated equipment, additive manufacturing has an entire engineering process preceding it consisting of model construction [3] generally using a virtual model built in a CAD software [6] or an acquisition of a physical model by a 3D scanner [7]. However, despite the complexity of the designing process, additive manufacturing techniques still have multiple advantages in several technical situations and applications in comparison to the multi-step conventional manufacturing methods [1], as synthesized in Figure 1.

It is important to mention that although AM is becoming more and more attractive and utilized in an expanding range of fields, there are still some areas in which traditional manufacturing exhibits major advantages compared to AM. Probably the most important consideration to be analyzed when choosing between these two routes is the production volume needed, which is a clear decisive factor, that consequently engages several other factors like additional time, total costs, and process global efficiency. The balance of all these factors in medium- to large-volume production is without a doubt ensured by the traditional manufacturing features, as the continuous production of a large number of parts will drastically diminish the cost per each part as well as amortize the initial high-cost investments. More than that, series production does not require design flexibility or product customization at any point after the production has started but rather requires a high resistance of the obtained products; therefore, the design benefits of AM are canceled in this situation. On the other hand, for low-volume production or customized products (such as individual-use cases that are crucial in patient-related medical areas), the issues associated with a large number of parts and less flexibility in the design become excrescent; therefore, AM features become major advantages in these situations. Besides the major issues related to time and costs, performance needs to be considered, in terms of materials and products’ properties. For AM, there is a smaller range of polymeric materials available, and most of them are thermoplastics with medium-range mechanical and thermal performance especially over repeated cycles of production and use, while the metallic alternatives generally require expensive and high energetic consumption equipment. In terms of product performance, when fine details together with complex functional design are required, the increase in AM parameters demands greatly diminishes the time-reduction advantages, with the risk of eventual fatal errors generating waste, and consequently, additional and unforeseen costs as well as time delays [8].

For both additive and traditional manufacturing, the features that represent major advantages in some application domains become major disadvantages in other domains and vice versa. Therefore, although the innovative and rapid benefits of AM could create the belief that this technique could replace traditional manufacturing, in reality, these classes of manufacturing techniques complement each other, so they cover all necessary industrial fields and respond to the specific needs of a growing number of applications and newly appearing situations, such as the COVID-19 crisis. Thus, an effective approach would be to consider AM’s unique features as an opportunity to cover areas where traditional manufacturing stumbles, or extend the use of traditional production processes, or combine AM with traditional techniques, where the product is manufactured using a hybrid technique (overprinting, over-molding).

Considering the great influence that the application specificity has on the selection of additive manufacturing as a production method, ASTM F42-Additive Manufacturing [9] introduced a classification of the AM processes into seven categories, according to the field that it is applied in, that are presented in Figure 2.

In close connection to the categories of AM identified, a classification of the main AM methods is presented in Table 1. 

Additive manufacturing usage is expanding continuously in the last decade; nowadays, these technologies are successfully implemented in a wide variety of industries that use concept models, functional models, patterns for investment and vacuum casting, medical models, and models for engineering analysis [6]. Therefore, besides the classification of AM techniques, the ASTM Committee F42 on Additive Manufacturing Technologies that is in charge of developing standards for additive manufacturing in a variety of industry-specific applications, settings, and conditions divided them by applications into 10 subsections: aviation, spaceflight, medical/biological, maritime, transport and heavy machinery, electronics, construction, oil/gas, consumer, and energy [10].

**Table 1 polymers-16-01055-t001:** Classification of the main AM methods [9,11,12,13,14,15].

AM Class	Materials Used	Principle	Techniques	Advantages	Disadvantages
Vat Photo Polymerization	Polymers (UV-curable photopolymer resins)	A liquid photopolymer in a vat is exposed to light source to be selectively cured into solid form	Stereolithography (SLA); Digital Light Processing (DLP); Continuous Liquid Interface Production (CLIP); Daylight Polymer Printing (DPP)	Rapid processing High quality finish of the part	High costs Extracting the 3D object from the mold generates issues
Material Jetting	Polymers (PP, HDPE, PS, PMMA), Waxes	Droplets of material are selectively deposited (jetted) on a substrate to build a 3D object	Material Jetting (MJ); Multi-jet Modeling (MJM); Nanoparticles Jetting (NPJ); Drop on Demand (DOD)	Less to zero waste	Difficult to apply in structural parts Post-processing required
Binder Jetting	Polymers (PA, ABS), Metals (stainless steel), Ceramics (Sand Glass)	Liquid bonding agent that acts as adhesive is selectively deposited to join materials in powder form	Powder Bed and Inkjet Head (PBIH); Plaster-based 3D Printing (PP)	Rapid processing No melting	Lower mechanical performance Post-processing required
Material Extrusion	Polymers (ABS, Polyamides, PC, PEI, PLA)	Thermoplastic polymer filament is extruded through a nozzle to build a 3D object	Fused Deposition Modeling (FDM); Fused Filament Fabrication (FFF)	Lower costs Good mechanical and structural properties High availability materials	Lower precision— many factors influence final model quality Accuracy and speed Nozzle requires technical attention
Sheet Lamination	Paper, Sheet Metals	Layers of material are joined together using an adhesive and printed one after the other (layer by layer) to build a 3D object	Laminated Object Manufacturing (LOM)	Low costs Acceptable accuracy	Limited material alternatives Post-processing required
Power Bed Fusion (PBF)	Metals (Stainless Steel, Aluminum, Titanium), Polymers (Polyamides)	Laser or electron beam melts or sinters the material in powder to build a 3D object	Selective Laser Sintering (SLS); Selective Laser Melting (SLM); Electron Beam Melting (EBM); Multi-Jet Fusion (MJF)	Suitable for prototyping Complex geometries	High costs Difficult to apply in structural parts Size limitations
Powder-fed Directed Energy Deposition (DED)	Metals (Stainless Steel, Aluminum, Titanium, etc.), Ceramics, Polymers	An electron beam, laser or arc energy source is directed toward a substrate material where it impinges with wire or powder feedstock material and melts, depositing the material on the substrate and building the part layer by layer	Wire Arc Additive Manufacturing (WAAM); Laser Metal Deposition (LMD); Laser Engineered Net Shaping (LENS); Laser Solid Forming (LSF); Directed Light Fabrication (DLF); 3D laser cladding	Suitable for repair/coat existing parts Machine large parts with high mechanical properties	Not suitable for small parts Lower detail accuracy and simple geometries

Where: PP—Polypropylene, HDPE—high density polyethylene, PS—polystyrene, PMMA—polymethyl methacrylate, PA—polyamide, ABS—acrylonitrile butadiene styrene, PC—polycarbonate, PEI—polyetherimides, PLA—polylactic acid.

Nowadays, additive manufacturing still generates concerns about the quality of produced objects, high process failure rate, and/or higher associated cost and time of the production process as opposed to traditional manufacturing [8]. As Sztorch et al. [8] emphasized in their study concerning the production of personal protection products required in the COVID-19 crisis, there are situations in which the functionality of the product becomes a decisive factor and the product is required to enter the market in a short period of time and to be produced in large quantities, as extraordinary situations from the COVID-19 pandemic proved. They compared the launch of traditional injection molding together with FFF to produce face shields, considering unit costs and production possibilities at various timeframes, responding to the emerging immediate need for the quick provision of personal protective equipment for medical services. The comparison showed that FFF printing needs to be optimized by increasing the process speed by 6–10 times concomitantly with increasing reproducibility and part quality and mechanical strength caused by interlayer defects that need to be reduced. Polyamide 6 helmets proved to be a viable alternative for rapidly launching the production of products by mold injection, while for 3D printing to compete with this, using large groups of printers could be considered an option, but 3D printing can cover the buffer period until traditional injection molding enters into production, which responds to the immediate and urgent crisis requirements by some niche fields.

Therefore, it becomes more and more clear that additive manufacturing and traditional manufacturing techniques as well as associated tooling are indeed not competitors, but rather complement each other in order to be able to respond successfully and efficiently to all the emerging requirements, technological evolution tendencies as well as exceptional situations, such as the ones generated by the pandemic years.

## 2. Additive Manufacturing Technologies That Use Polymers

The rapid evolution of additive manufacturing techniques adapted for polymeric composites development has evolved together with the circular economy growth and need for sustainability progress. As additive manufacturing of polymeric products and tools successfully supports the recycling and reusing of waste and/or used products to reintegrate them in a process chain, it greatly contributes to the circular economy concepts related to reducing raw-material consumption, waste, energetic consumption as well as costs related to manufacturing [16].

Since the birth of additive manufacturing, a multitude of methods have been introduced, customized, and personalized in several applications, from medicine, biomedicine, and tissue engineering to architectural design, automotive, aeronautics, and aerospace [17]. For plastic-based 3D-shaped products or tools, most studies and companies use material extrusion and vat photopolymerization, as they both allow the integration of reinforcing fibers into the polymer, and thus develop 3D-printed polymeric composites [3]. Material extrusion uses Fused Deposition Modeling and/or Fused Filament Fabrication techniques. The FDM additive manufacturing method was patented by Stratasys company in 1989 [18]; the term was trademarked in 1991 [19]. The term “Fused Filament Fabrication” began to be used when referring to other devices than that patented by Stratasys which used the same principles, in order to avoid litigation for copying their “FDM” trademark. Technically, both terms describe the same principle.

The three most used 3D printing techniques of plastic materials are fused deposition modeling, stereolithography, and selective laser sintering [20]. FFF/FDM and SLS use thermoplastics, while SLA uses thermosets; each of the techniques is presented below.

### 2.1. Fused Deposition Modeling

Fused filament fabrication is one of the most common techniques for polymer-based AM being widely used for printing components (from prototypes to functional end-use parts) manufactured from thermoplastic polymers. As described in Table 1, fused deposition modeling uses thermoplastic filaments as extrusion materials. As Figure 3 illustrates, the filament is subjected to heating until it reaches a molten state and extruded via the rollers rotating in opposite directions, through the nozzle of the printer, which moves in three degrees of freedom and deposits the polymer on a platform, building the part layer by layer according to the instructions and coordinates given through the design software-generated file [21].

In general, the consumer-level FFF/FDM technique ensures lower resolution and accuracy compared to other 3D printing processes using plastics, these two features being greatly influenced by the thermoplastic filament properties and the parts generally needing surface post-processing (i.e., chemical or mechanical polishing). Also, during the deposition of the layers, the formation of voids between them is a common problem which imprints a high closed porosity to the parts, influencing their capacity to bear mechanical loads. Therefore, this technique using consumer-level equipment is generally not suitable for complex designs or highly detailed parts, but it is a very attractive alternative for hobbies, DIY (Do-it-Yourself), and basic laboratory research studies helping students, researchers, engineers, and technicians. When using this technique on an industrial level, the available equipment provides some solutions to the drawbacks, and a larger variety of thermoplastics and even composites, but the price is commensurate with all the extra features [20,21].

The mechanical properties of components that are produced by FFF depend on the printing parameters, which are optimized to maximize the part quality, the microstructure, and the overall printing process economy [22,23]. FFF/FDM is currently confidently used in space hardware manufacturing applications for launch vehicles and spacecrafts [23]. FFF/FDM use a wide range of thermoplastics, from engineering nylons, ABS, and PLA to polyphenylene sulfide (PPS), polyetherimides reinforced with different fillers or blended with polycarbonate (known as ULTEM 9085 [24]), glycol-added polyethylene terephthalate (PET-G), thermoplastic polyurethanes (TPU), and high-tech thermoplastic consisting of polyetheretherketone (PEEK) and polyetherketoneketone (PEKK) [25]. The different classes of materials available to be 3D printed via FDM each possess specific advantages by their unique properties including transparency, biocompatibility, FST (flame–smoke–toxicity) certification, chemical resistance, heat resistance and strength, durability, etc., facilitating the material selection according to the target application [25].

Table 2 presents a summary of some of the most widely used polymers for 3D printing via FDM processing, showing their advantages and disadvantages when used in AM, together with their major fields of application.

Besides the basic thermoplastic solutions for printing 3D parts, innovations in technology and materials have expanded their unique properties and usage by adding different compounds into the polymer and strongly enhancing the final products’ performance and capacities [18]. There are applications that use FDM-printed parts from filaments infused with metallic, glass or ceramic compounds, in which the polymers are melted away by debinding and sintering to produce robust materials for electronics [18].

Amongst the tailored polymers for 3D printing are ULTEM materials, developed by Stratasys, ULTEM 9085 [24] being widely used for space applications as it offers high thermal stability, flame-retardant performance, chemical resistance, and high specific strength [38]. Tailoring of ULTEM properties for the improvement of its performance is presented in several research papers; most of them focus on improving its mechanical properties by the modification of printing parameters (i.e., layer thickness, raster angle and width, chamber temperature, print orientation, etc.) as well as the limitation of water uptake [39,40,41], the strength variation range being between 45–85% compared to injected parts.

It is clear that polymers can be successfully processed via AM methods, using them in a multitude of forms and compositions, from single polymer and polymeric blends, micro and nano composites, to short and long fiber-reinforced polymer composites, the used technique depending on the chosen compounds’ processability features and target application. In the past decade, notable progress has been made in the field of 3D printing polymeric composites reinforced by fibers; considering the unique properties of polymers combined with the enhancements ensured by fiber-reinforcing agents, the immense benefits provided to the additive manufacturing sectors are of great value [42]. In the present, Stratasys manufactures FDM filaments composed of Nylon 12 and carbon fibers to produce parts as strong as aluminum, allowing the replacement of metal in different applications, exhibiting the highest flexural strength of any FDM thermoplastic, which leads to the highest stiffness-to-weight ratio [43]. However, for the development of molds, mechanical stresses are only some of the factors that influence their viability in different applications; therefore, strengthening with the aid of fibers is still limited to specific uses.

### 2.2. Stereolithography

Stereolithography was the first 3D printing technology and it remained until nowadays one of the most widely used for professional applications, due to the highest resolution and accuracy, high level of details, and high-quality surface finish that requires no further processing. Due to the high precision of the technology and chemical bonding formation between layers, the resulting parts are isotropic, and their mechanical performance is not influenced by process parameter variation. Given all these factors, the technique is optimum for highly detailed prototypes, such as molds, functional parts, patterns, jigs and fixtures, jewelry, dental implants, and end-use parts [20].

SLA belongs to the VAT polymerization class of AM techniques. According to Figure 4, a liquid, photosensitive thermoset resin is poured into a vat (tank) and interacts with a UV light for selective polymerization, the UV light curing the resin layer by layer until the final part is obtained. In SLA, layer thickness (or height) is generally approximately 50 µm but it can reach 10 µm, when extremely high quality is required, and time allows it.

Besides all these advantages, the SLA technique uses a wide and versatile resin formulation, covering several properties of tailoring (optical, mechanical, thermal, biocompatible). The materials’ availability and properties are strongly dependent on the manufacturer and associated printer equipment.

The most important advantages of all SLA resins are high stiffness, highly smoothness of surface, and fine and high-level details, while the most important disadvantages are their low elongation at break that leads to brittle fracture, susceptibility to creep and UV radiation that affects their properties over time in outdoor applications [44]. Available SLA resins exhibit properties similar to some thermoplastics (i.e., ABS, PC, PP, etc.), being heat resistant by their high heat deflection temperatures (HDT), hard, flexible, impact-resistant, biocompatible or transparent, depending on their type [45]. Table 3 presents some of the main types of SLA resins, generally available at most manufacturers in different registered tradenames. The most commonly commercial SLA resins are manufactured by Formlabs, Protolabs, 3DLite, etc., each offering its own customized range of products for this application. Amongst them, Formlabs offers the most comprehensive resin library with over 40 SLA 3D printing material alternatives. In addition, the of the main types presented, Formlabs offers additional SLA resin alternatives such as flame-resistant resins (designed for indoor and industrial environments with high temperatures or ignition sources, like interior parts in aircrafts, cars, trains, protective and internal consumer/medical electronics components), Silicone 40A resins (first accessible 100% silicone 3D printing material with superior properties of cast silicone suitable for small batches of silicone parts, customized medical devices, flexible fixtures, masking tools, soft molds for casting urethane or resin), draft resins (up to 4 times faster than standard resins and 10 times faster than FDM), polyurethane resin (excellent long-term durability, stability to UV, temperature, humidity, flame retardancy, chemical and abrasion resistance, sterilizability), resins for medical and dental parts (biocompatible resins for producing medical and dental appliances), and jewelry (for easy investment casting and vulcanized rubber molding, with intricate details and strong shape retention) [20].

SLA is known for creating high-resolution parts with good surface finish, but tensile strength can sometimes be affected; therefore, the careful choice of the material used is an important parameter for this technique as well [45].

### 2.3. Selective Laser Sintering

The SLS 3D printing technique belongs to the powder bed fusion AM class, which is generally applied for metals but can be applied for polyamides and a few other thermoplastics within the polymeric materials class. Its ability to produce strong functional parts at a high productivity rate generating low costs per part makes this technique trusted in a wide range of industries for applications such as rapid prototyping, manufacturing aids, low volume or custom production [20].

SLS 3D printing uses a high-power laser to sinter small particles of polymer powder into a solid structure based on a 3D model, as illustrated in Figure 5. The printing process develops over three main stages: (1) preheating—during which the powder bed is heated to a predefined temperature (bed temperature just below the softening temperature of the polymer that is used to minimize the laser energy and eliminate any distortion of the piece during cooling), held constant throughout the part-building process; (2) building phase—core phase of the fabrication process involving several operations (the lowering of the platform to receive the powder particles dragged by the roller or by the spreading blade, laser beam melting of the layer of particles along a computerized trajectory, gradually cooling down to the bed temperature for solidification); (3) cooling phase—during which the heat source is switched off and the powder bed cools, gradually cooling until it reaches the extraction temperature of the piece [46]. The unfused powder supports the printed part during the process, so it eliminates the need for dedicated support.

SLS is an optimum choice for the printing of complex geometries, generating isotropic structures, and although surface finish is rather rough, post-processing is easy. As mentioned, compared to FFF/FDM and SLA, the available materials for SLS are very limited (mainly polyamides, sometimes PP, flexible TPU, TPE, and more recently, PEEK and PEKK), but the small class exhibits excellent mechanical performance, similar to injected parts [47], as presented in Table 4.

SLS is one of the 3D printing techniques that generates parts with one of the most isotropic compositions, and together with the use of high strength plastics, results in high-performance products.

Comparing the three 3D printing processes that use polymers, each of them can be a choice for different applications and requirements, as each of them has its own advantages as well as disadvantages. FF/FDM offers low-cost consumer equipment alternatives and uses widely available materials, SLA offers high accuracy, precision, and a smooth surface finish using a large variety of functional materials, while SLS ensures strong functional parts, without support structures during printing, and freedom of design. On the other hand, each of the three techniques exhibit drawbacks that could make them unsuitable for some applications. FDM only provides low accuracy and detail level with limited design when consumer equipment is used, with professional equipment mitigating some of the drawbacks but coming at a high cost. Parts that can be printed with SLA materials are often sensitive to long-term exposure to UV light, making them generally inaccessible for outdoor applications. SLS can be used with a limited range of materials and hardware equipment costs are higher. The selection between the three technologies needs to take into consideration all these aspects in the context of the cost investments, sustainability, application requirements, and available materials and equipment.

## 3. Technologies That Use Molds

Molds represents one of the most used tools in the manufacturing industry with applications in several fields, from consumer goods to sports, medical, transport, and security. In the high competitiveness encountered in the mold and tooling industry nowadays, the time needed for a product to reach the market (time-to-market) represents an important factor to be considered by the companies, along with the quality of both the product and the mold, when building their tooling for development of products.

In today’s competitive mold industry, a product’s time-to-market plays an important role in the success of a company producing quality molds [50,51].

The molds and dies industry is the root of the manufacturing world, as they represent key elements in mass production. Molds find extended use in a wide range of technological processes, especially related to plastics (casting, injection, extrusion, compression, blow, rotational molding, resin transfer, etc.). Dies are mostly associated with metals, being implicated in technological processes like stamping, forming, metal injection, etc.

However, although tools are crucial for worldwide industrial fields, the molds industry faces some challenges as digital tooling expands. First, it is a capital-intensive industry in which the manufacturing costs are consistently increasing but the price of mold and die increase does not occur at the same rate, thus threatening the survival of competitors, automated shops, and factories which imply a decrease in human personnel and a lack of trained personnel to operate the machines [52]. Last but not least, the materials used for most of the industrial molds are expensive metallic ones, which require long timeframes and expensive manufacturing as well as secondary preparation stages; these factors represent unbalanced investment when low-volume production is needed or in application where the manufactured parts need constant tailoring and customization.

Therefore, considering all these aspects in the global economic and technological circumstances, it becomes urgent to direct the mold industry towards optimizing costs, improved efficiency, and advanced forecasting. There are a series of promoters that can significantly contribute to the aligning and allowing of the growth of the mold industry in the current economical/technological worldwide trend, some of them being 3D printing for prototyping, 5-Axis CNC precision machining, rapid tooling systems, and advanced CAM/CAD tools [52].

The main technologies that use molds as a main tooling method are injection molding, melt compounding, vacuum bagging liquid injection molding, casting, thermoforming, and composite fabrication, as well as different specific applications such as dedicated/customized biomedical devices.

These molding processes imply the use of different mold components, depending on the part targeted to be manufactured as well as the polymer type used. Thermoplastics can be molded by melting followed by cooling, while thermoset can be molded into different shapes by pouring/laying-up of resins in a liquid state and curing (at room or high temperatures). Some of the most important molding processes are listed below [53]:Casting—it is the simplest molding process, as it requires simple tooling and low costs, and can be performed at low pressures. The thermoplastic is heated until it reaches a molten state, poured into the mold, and allowed to cool before extraction from the mold. Although it allows the production of complex shapes, it can be used for parts with a thickness higher than 12–13 mm.Injection molding—it is one of the most extensively used techniques for molding plastics or metals as it allows the production of three-dimensional parts which can be easily reproduced. The material brought in liquid form is inserted/injected at a high pressure into a closed, cooled mold, filling it and taking its shape. The molded material is extracted after complete cooling and solidification. It is a process suitable for large quantity production (i.e., more than 30,000 parts per year). Despite the use of expensive tooling (i.e., expensive metallic molds), the large volume production ensures its cost-effectiveness; however, recent trends promote its use for smaller production volumes with the tooling adaption.Extrusion molding—it is similar to injection molding, but with the difference that the molten material is inserted/injected through a die and the obtained structure is linear and rod-like (not necessarily cylindrical). After cooling, the rod structure can be cut at different lengths depending on necessities.Compression molding—it is the most complicated molding process, in terms of labor, being used only for large-scale production (such as a higher number of small parts in boats, the automotive industry, etc.), and not for mass production. The liquid molten material is poured into a lower mold and compressed with an upper mold into the desired shape and extracted after complete cooling and solidification. The high temperatures used ensure material strength.Blow molding—it is a process mainly used for pipes and milk bottle production, allowing the production of up to 1400 parts in a 12 h shift, with uniform wall thickness achievement. Although it uses the standard concept, it requires several different mold parts. The plastic in a melted state is injected into a cold mold, concomitant with air blowing into an attached tube, pressing the plastic against the walls of the mold so that it takes the shape of the mold. After complete cooling, the part is extracted.Rotational molding—it is an environmentally compatible process, as raw material does not go to waste. The process involves high-speed rotating using two mechanical arms, the mold that contains the hot liquid material, which uniformly coats the mold surface, and the final part has a uniform wall thickness and hollow shape. It is widely used for toys, tanks, and different other consumer goods.

The minimal requirements of 3D-printed molds come from the requirements imposed by the molding technology used together with the molded material properties. The most important requirements that a 3D-printed material has to respond to so that it can be used as mold tooling are referred to in [54]:Suitable mechanical properties, especially in terms of high stiffness—for example, injection molds must exhibit suitable mechanical performance to withstand the high pressure used during injection while maintaining a good dimensional stability (no deformation) and accuracy over multiple-use cycles.Suitable thermo-mechanical properties, in terms of resistance to high temperatures without showing deformation, meaning that the polymer used needs to exhibit a high value of heat deflection temperature in order to ensure a precise control of the process and the required dimensional stability.Dimensional accuracy is crucial for the production of parts with a high level of details.

Considering the limitations that polymers have by their own physico-chemical nature in the context of the materials requirements for molds and molded parts, using 3D-printed molds, especially for the injection molding technologies, narrows down their beneficial use to some technological situations such as referenced in [54]:When fast turnaround times are needed (1–2 weeks for 3D-printed molds as opposed to 5–7 weeks for traditional ones);Low-volume production (applications where a maximum number of 50–100 parts are needed);Small-size parts are to be produced (up to a maximum of 150 mm);Applications where design changes or iterations are foreseen.

The two 3D printing processes that can produce parts with high accuracy and smooth surfaces without requiring complex post-processing are material jetting and SLA. Materials jetting is used exclusively on an industrial scale, while SLA is available on both an industrial and a consumer level, although the materials and capabilities cannot be considered for high-end production [54].

Considering that in molding processes, the final part quality is greatly influenced by the mold features, there are a number of factors to be taken into consideration in terms of design to obtain the desired product quality via the desired process efficiency [55]:Selection of optimum material—the used materials need to withstand the parameters required to be implemented during the molding process (i.e., temperature, pressure) without melting, warping or deforming.Design considerations—the design of the mold needs to be optimized to build molds for any molding processes, especially injection molding, as design items (i.e., number of walls, wall thickness, draft angles, infill patterns, etc.) generate significant modifications to the quality and durability of the mold and consequently to the quality of the part and cost investments in the technology for the product.Testing trials and validation stages—as with any product or other processes, molds printed via 3D need to be tested in terms of resistance to the conditions required by the parameters used (thermal resistance, mechanical resistance and dimensional stability at the processing temperatures, pressures generating mechanical loads and during a required number of cycles), in order to establish the molds’ limitations and perform adjustments if needed, before production starts.Production volume considerations—especially for injection molding that generally is suitable for thousands of cycles, 3D-printed molds cannot surpass traditional metallic tooling and can only be used when low-volume production (50–100) is possible due to the modification of their properties after a number of cycles; therefore, they can only be used in rapid prototyping, low-volume production, and other molding techniques that require single use or constant tailoring of the design.Size and shape of the molds—the selection of the mold type needs to take into consideration that it has to handle the size of the part to be manufactured, as generally mold machines by CNC are larger, and molds produced by 3D printing exhibit some size limitations compared to them.Surface finish—considering the high degree of surface finish offered by metallic molds (aluminum or steel), 3D-printed molds tend to exhibit generally rougher surfaces, decreasing the surface finish quality, without taking into consideration the degradation scenarios during injection molding, for example, rendering them the less suitable candidate in some applications.Draft angle—this factor needs to be considered especially for injection molding and composite fabrication, as its correct selection can contribute significantly to the facile extraction/demolding of the part at the end of the process.

As 3D printing technologies are in constant development and improving dynamics, the use of tooling produced by additive manufacturing techniques continues to expand. However, although AM appears to be replacing traditional tooling manufacturing, in reality, these two classes of techniques are partners rather than competitors at the risk of eliminating one another. Traditional manufacturing exhibits some clear benefits that could not be ensured (at least in the near future) by the AM alternatives, such as the fact that it allows high-volume production with lines that can run for 24 h daily, reduces cost-per-unit due to amortization of upfront tooling costs, and provides strong part consistency due to the possibility of repeating the same manufacturing cycles without deviation from the original design intended [56]. The major drawbacks of traditional manufacturing represented by the high waste of materials, inflexibility of the original design tailoring, and high costs for production quantity below a large volume (medium to low to single-use) are actually the major benefits of AM in their reversed form. Therefore, additive manufacturing emerges as a solution for the fields where the major drawbacks of traditional manufacturing generate a high level of impediments and disadvantages.

## 4. Applications That Use 3D-Printed Polymeric Molds

As already mentioned, there are sectors and applications in which traditional manufacturing techniques and their additional tooling cause significant technological and economic issues. Therefore, the use of AM tooling can optimize the supply chain and productivity by allowing the advanced and rapid customization of products, with improved functionality and weight implying reduced lead times and costs. AM tooling manufactured with polymeric materials is particularly useful in the low-volume production of high-complexity parts, where reiteration of design is a major requirement allowing the functionality improvement of the final product, and in the cases where weight reduction and fast lead times of the tooling are an advantage [57].

### 4.1. 3D Printing of Molds for Injection Techniques

Injection molding is one of the most established and important processes for mass production of objects and products from thermoplastics, usually without the need for additional finishing [58], being the second technology in plastic industry production, after extrusion technologies [59]. Injection molding is a mass production process as it allows the manufacturing of a large series of the same product with high quality [60,61].

One of the most important drawbacks of injection molding manufacturing is the high costs and extended lead time for designing and procuring the molds [62]. Industrially, the most commonly used materials for molds manufacturing are steels and aluminum, considering their machinability, variety in composition and properties, heat treatment possibility, higher thermal conductivity, and the ability to be coated for improved surface finish and good polishing ability [63,64]. Molds that are manufactured for high-volume production (up to millions of parts) require extreme durability and hardness/toughness and have to maintain their dimensional stability and special properties for thousands of thermal cycles [65]; therefore, steel alloys are the chosen material solution. However, although these harder materials exhibit all the positive properties, their requirement for special tooling imposed by the higher effort needed to mill them leads to a significant cost increase [66]. Therefore, for lower volume production, the higher costs exhibited by steel alloys machining into molds (generally achieved via CNC or electrical discharge machining techniques [62]) would increase the technology expenses beyond economic effectiveness. Additive manufacturing can bring important optimization advantages when combined with formative manufacturing such as injection molding, due to the advances in machines’ design and materials [67]. Combining 3D printing for mold tooling seems to ensure a more cost-effective route compared to traditional metallic molds. Recently, plastics companies have shown an interest in using AM to manufacture molds for injection molding which can be used to produce end components in low-volume production; however, until now, there is no clear indication whether these parts are brought to market as independent products or components of a product, nor is there any indication of the cycle life of a mold produced using AM [68]. Rapid tooling is a term that describes the use of AM to achieve molds ensuring shorter lead times compared to conventional techniques [69]. Three-dimensional printing is a powerful solution for fabricating injection molds rapidly, with high flexibility and involving low costs, as it requires limited equipment, saving valuable CNC time and skilled operators for other high-value tasks. Molds manufactured via AM techniques can be obtained from industrial machines as well as from small-size laboratory equipment allowing design testing and iteration at a lower scale before investing in expensive tooling for mass production [62]; this diminishes raw material consumption during trials and non-profitable investment risks.

When choosing between 3D-printed and traditional tooling for injection molding processes, the volume of production is a crucial factor to be taken into consideration, as the features of AM can pass from advantages to disadvantages when large-volume production is used. Therefore, the additive manufacturing features of fast launching of the concept, high versatility and flexibility towards corrections required at almost any point during production, achieving high complexity geometries without significant increase in costs, and time and cost-effectiveness [64] represent major advantages when they are used as tooling methods instead of traditional ones, in low-volume, single-use or highly customizable case production.

Stratasys [70] summarizes a comparison between different methods to produce prototypes via injection molding, in terms of the number of parts, materials, average mold cost and average cost/past, which is presented in Table 5. However, Stratasys [70] mentions the important observation that with the use of FDM methods, the mechanical properties of the developed thermoplastic prototypes are not comparable with the ones obtained via traditional injection molding, as both processes and injected materials are different. Also, besides this, when producing a medium to a large volume of parts, although the time, cost, and post-processing required to produce the injection mold are significant, the long operation lifetime and large number of parts able to be manufactured compensate the investment to such an extent that it fully amortizes.

Considering all these aspects, polymeric materials are the most appropriate candidates for low-volume production of molds via additive manufacturing. Special care needs to be given to the properties of the polymer used for the mold versus the polymer to be injected into the mold, as the choice for the mold needs to present melting temperature above the one exhibited by the polymer used for injection. In this sense, there are companies that developed molds via 3D printing, as well as research studies investigating these technical alternatives. The company Formlabs manufactures polymeric molds via 3D printing stereolithography using their customized photo-curable resins, suitable to replace aluminum molds in injection molding applications for low-volume manufacturing, with cost reductions that could reach 80–90% and time reduction by 90% [73]. Depending on the necessities, Formlabs provides a large range of resins, each of them having one or more advantages such as high molding temperature and pressure properties/increased number of operation cycles/wall thickness/reduced costs [73]. However, their available classes of resins exhibit heat deflection temperature values up to a maximum of 238 °C [73], limiting their use with the injected materials to polymers with thermal resistance below this temperature, such as commodity thermoplastics (i.e., PLA, PE, PP, PS). Formlabs often applies encapsulation of the plastic molds into aluminum frames for better pressure withstanding and preventing warping and deformation after several thermal cycles [64]. Figure 6 illustrates the major steps in the workflow of the injection molding process when using 3D-printed molds.

Stratasys also manufactures molds from their customized ABS photo-curable resin, but using the PolyJet technique, in which the resin is jetted and UV-cured [74], provided with additional cooling systems to maintain the temperature below 58 °C when injecting ABS. Using Stratasys printers, the Canadian Javelin company provided 3D-printed ABS molds for injection applications, attesting that their products ensure a 50–70% cost reduction compared to aluminum tools, while maintaining the same advantages offered by the metal molds [75]. Meanwhile, Protolabs compared three manufacturing routes of a custom plastic fitting for a motor housing: injection molding ABS parts with SLA 3D-printed molds, industrial FDM ABS 3D printing, and traditional injection-molded ABS part [72], with their summary recording that the first method reduced the lead time by four times and the costs by almost 24 times compared to the traditional one. Their study concluded that 3D printing the injection molds is the most cost-effective way for low-run injection molding and that material jetting and SLA are the most suited technologies for 3D printing injection molds, while the lifetime of molds can be improved by using some technical issues (i.e., using wide draft angles, release compounds, keeping the part volume below 165 cm^3^) [72]. They attested that depending on the material injected, 3D-printed molds could be used for cycles between 30–100 runs [72].

Dizon et al. [67] investigated the possibility of using different 3D printing methods and materials to manufacture polymeric molds for injection molding applications, starting from Formlabs and Stratasys materials and techniques. Injection molds having a chosen geometry (in their case, a cube) were printed via three different techniques: stereolithography—using the Formlabs printer and resin, PolyJet—using the Stratasys printer and materials, and Fused Filament Fabrication—using the Intamsys printer and Evonik PEEK material. For the molds manufactured by SLA and PolyJet made of photoreactive methacrylate-based resin and ABS-based material, respectively, excellent finishes were acquired, but for the ones manufactured via FFF from PEEK, the structure delaminated after the process. Good dimensional accuracy of injected parts manufactured from PLA material was achieved using molds manufactured by SLA and PolyJet printing.

As with the 3D printing market, stakeholders are facing more and more competition; several companies and research laboratories have extended their applications to 3D-printed molds for prototype part production, shortening research and development activities time to 35% and reducing costs up to 90% [64]. In addition to this, applicative research has attested that using 3D-printed molds allows an incomparable flexibility in terms of geometry and design.

Chung et al. [76] have conceived a method for rapid and low-cost production of liquid elastomer injection-molded devices that utilizes fused deposition modeling 3D printers for mold design, enabling rapid prototyping of elastomer devices with complex geometries and requirements, which is a hallmark of fields such as production of medical devices. The authors created the mold from ABS material via fused modeling deposition, taking into consideration surface smoothing for fine-tuning the mold by oversizing the mold (adding extra material) and sanding to desired dimensions or treating ABS with acetone for gradual dissolution. The low costs and reduced production time allow for several iterations of the design that allow corrections or modifications according to the device geometry. However, when using ABS material for molds, the low heat deflection temperature (90–100 °C) needs to be taken into consideration as it lowers the operating temperature down to 70 °C, limiting the range of polymers that can be injected. If higher curing temperatures are needed, other FDM/FFF suitable materials can be taken into consideration, such as polycarbonate. An ABS mold lasts around 20 uses before the ABS plastic wears out, cracks, or suffers damage due to compressional stresses and heating cycles [76], and generally FDM/FFF-printed molds can be used in direct rapid tooling for the limited number of shots in injection molding [64].

In 2018, Altaf et al. [77] conducted a study during which parts made by ABS and nylon mold inserts printed by the FDM technique showed a good performance, comparable to the machine metal mold, for a small number of metal injection molding cycles, concluding that enhanced polymer mold inserts could be a feasible choice in this process for low-volume part production, prototype manufacturing, design validation, form and fit analysis, and other upstream processes, prior to permanent mold manufacturing.

Depending on the material used, geometries and sizes of injection molded parts, as well as the additive manufacturing route used, Stratasys [70] attests that 3D-printed molds can withstand producing from a few dozen to dozens of thousands of parts. The ideal required molding temperature should not exceed 250–300 °C; therefore, polymers with melting/molding temperature higher than 250 °C or that exhibit high viscosity in the processing temperature domain will generate issues regarding the final products’ quality, and they will shorten the mold life. Depending on the parts’ geometries, size, and complexity, and most importantly, the class of material injected, molds’ lifespan can vary from a few dozens to tens of thousands of cycles. Generally, traditional molds withstand more than 10,000 cycles with any polymeric-based material injected, while the metal laser-sintered ones can reach this number only when injecting standard polyolefins, PS, ABS or thermoplastic elastomers, the number decreasing below 100 parts when injecting fiber glass-reinforced PC or PA, PPS or PPO (polyphenylene oxide) polymers. When injecting products in cast resin manufactured molds, the standard thermoplastics can be formed using the same mold up to hundreds of cycles, and only a few dozen when injecting fiber glass-reinforced PC or PA, PPS or PPO. PolyJet molds can be used to produce standard thermoplastics (polyolefins, PS, ABS or thermoplastic elastomers) in an average number of 200–300 parts and a few dozen parts when injecting plastics like acetals, PC/ABS, and glass fiber-reinforced PP [70].

Godec et al. [78] studied the AM PolyJet process and its possible application for the production of bridge polymer molds for injection molding of a small quantity of the molded parts together with design rules for PolyJet bridge molds, dividing 3D-printed molds into three categories, depending on the durability [78]:soft (temporary) tool/molds (i.e., silicone molds)—as expected, they can be used for a very limited number of cycles before they reach their usage period.bridge tool/molds (i.e., plastic molds)—can be used for small-batch production (i.e., hundreds to thousands) and they generally require shorter manufacturing periods, their durability being strongly influenced by the material used for production within them.hard tool/molds (i.e., metallic molds)—can be used for large-batch production (i.e., hundred thousand), similar to the molds manufactured by conventional methods, but they require longer processing time and costs, compared to the other two categories.

In their studies, Godec et al. [78,79] attest that PolyJet molds are not intended to be designed to replace soft or hard tools used in medium- and high-volume production, their purpose being to fill the gap between them and sometimes act as substitutes for 3D-printed prototypes. Although the major advantages are a short time for manufacturing and printing at room temperature, successful injection molding using these molds requires taking into consideration additional factors such (such as design, manufacturing, and post-processing).

Another study [80] focused on comparing different AM technologies (SLA, Laser Sintering, and PolyJet) with different additive manufacturing polymers. The PolyJet resin had similar behavior and properties to ABS and the mold was tested to inject elastomeric polyethylene (injection temperature of 95 °C), polypropylene (injection temperature of 200 °C), and ABS (injection temperature of 270 °C). The mold withstood before cracking to a total of 20 parts (6 PE, 8 PP, and 6 ABS). For the LS method, the mold was produced from polyamide 12 filled with 50% Al, and for the SLA, the materials used were tough resins and high-temperature resins filled with 1/5% carbon nanotubes and graphene nanoplate. The SLA-produced mold with tough resin suffered from warping after printing and UV curing, producing a curved surface that makes this mold unusable for the injection of plastic materials. The laser-sintered mold with PA50Al had the lowest surface definition (detail finish), as expected, whereas the 3D-printed mold with ABS-like resin had the highest surface definition. The manufactured molds were successfully validated for short series productions and for obtaining final parts ready for product validation by using conventional polymers as PP and technical polymers as ABS. Also using UV-curable resin, but this time an acrylate-based one, Noble et al. [81] used an inkjet 3D printer to develop molds for the injection of parts for artificial photosynthesis device prototypes. The results were promising; although the directly 3D-printed parts did not have adequate surface finish for molding optical components, the surface finishing treatment (steel-shaft hot pressing, printer resin coating, scrapper and buffer polishing) tested afterwards added improvement to the final samples.

Also, in medical devices when higher resolutions are needed, FFF methods can be replaced by SLA, if costs justify it. SLA is commonly used for prototyping and low-volume runs of polyurethane devices by printing a mold master and casting a silicone mold around the mold master to create the mold for polyurethane injection [76].

Although considered in the last decade as problematic materials from a sustainability and circularity point of view, epoxy resins were also studied as candidates for injection molds manufacturing [82]. Rahmati and Dickens [83] produced SLA injection molds using SL epoxy that was successfully used to inject 500 PP and ABS parts. The molds’ failure was caused by mechanical loadings in flexural or shear during the injection process, as the temperature of the epoxy molds was reduced to 45 °C before each new cycle.

The studies performed by researchers and small-scale manufacturing companies tend to attest that SLA-printed molds are generally feasible in replacing the expensive metallic molds needed when producing medium-scale quantities with low melting temperature thermoplastics such as PLA in a small-scale production facility.

Besides the widely known thermoplastic polymers processed by injection molding, depending on their customization requirements, rubber molded products are also processed via injection molding (organic rubber molding, Liquid Injection Molding or Thermoplastic Rubber Injection) [84]. Structur3d, a developer of soft materials for additive manufacturing builds, customized water-soluble PVA molds for use in custom-manufacturing rubber parts through liquid injection molding. Their solution consisting of sacrificial dissolvable 3D-printed molds allows the manufacturing of fine and complex design parts while removing the drawbacks associated with damaging the rubber parts during extraction from the molds. Moreover, sustainability is addressed, as the 3D-printed PVA molds exhibit suitable thermal stability to withstand rubber processing temperatures, while being 100% biodegradable, non-hazardous compounds that generate no hazardous by-products during removal by water dissolution [85].

Besides the mold material selection and surface finish treatments, the mold design greatly influences the time and costs invested in manufacturing and using the molds (whether polymeric or metallic) in injection molding. The cooling system choice of injection molding tools is an important factor that greatly influences the total production time, as the cooling stage represents about half of the overall production cycle [86] and cooling temperature, speed, and time generate strong effects on the injected polymer crystallization kinetics [87].

Some of the earliest research studies involving 3D-printed mold cooling systems design were performed in the early 2000s. Sachs et al. [88] compared surface temperature achieved using 3D-printed molds with conformal channels and machined molds made of stainless steel with straight channels, concluding that the printed ones exhibited a more uniform surface temperature. Xu et al. [89] demonstrated simultaneous improvements achieved with 3D-printed tools with conformal cooling channels in terms of production rate and part quality as compared with conventional production tools.

Since the inception of research on the topic, several studies have been conducted [90,91,92,93]; the subject still remains a challenge nowadays, as optimum configurations are still discussed. Injection molding tools with conformal cooling channels can only be achieved by additive manufactured molds, traditional die design being limited to straight drilled cooling channels. Jahan and El-Mounayri [94] recently proposed a methodology to determine the optimum design of conformal cooling channels in injection molds, their results showing that for different plastic part designs, different channel configurations provide optimum solutions when taking into consideration cross-section dimensions, section size, pitch distance, and mold wall to channel centerline distance. Their study provides a guideline for an easier selection of conformal channels’ design parameters.

Besides the improvement of the thermo-mechanical performance of 3D-printed materials for injection molds requirements considering structural design and geometries, literature attests to significant opportunities in terms of the research of polymers, composites, and nanocomposites to enable rapid tooling with toughened materials via 3D printing techniques [64]. Considering injection molds, the need for toughened high-performance polymer-based materials in terms of thermo-mechanical properties and behavior lead to intense research on improving the fracture toughness, delamination, thermal properties, and heat transfer. All these items could be achieved to a high extent with the use of 3D printing methods to develop improved semi-crystalline thermoplastics as well as thermoset, in formulations that involve nanoparticles addition [95,96]. Addition of graphene oxide nanoparticles to 3D-printed TPU/PLA [95] led to high-quality complex shape nanocomposites parts with improved crystalline structure, 90 °C lower degradation temperature, and approximately 170% higher compression modulus and 75% higher tensile modulus. Besides graphene oxide, carbon nanotubes, nanoclays, nanosilica, and nanocellulose are the most commonly used nanofillers added to 3D-printed materials [96]. Still, the use of expensive nanofillers in applications destined for short lifecycle injection molds needs to be very well balanced in terms of performance versus costs evaluation.

Besides the growing application of 3D-printed molds for injection molding of polymers, research studies [97] extend injection molding to ceramic feedstocks from Al_2_O_3_ and MoSi_2_ containing composite to produce a variety of parts with demanding geometries such as spirals, cages, and helices. Sacrificial molds from PLA were 3D printed via FDM and compared with DLP-printed ones, from water-soluble resin based on Polyvinylpyrrolidone, showing that the latter one is more suitable for the high resolution required by the products with small structural features. Although sacrificial, these molds imply costs smaller than 10 USD and production time in days of magnitude, compared to the traditional steel ones that can costs from 10,000 to 100,000 USD depending on their complexity and require production time from weeks to months.

### 4.2. 3D Printing of Molds for Casting Techniques

In casting, a hollow mold is created from a master, which can be hand-sculpted or more recently 3D printed, that is afterwards immersed in a casting material (i.e., sand, clay, concrete, epoxy, plaster, silicone) that hardens. Plastic or metal is poured into the mold, and the master is either removed or burnt out to create the final part [98]. Metal casting is widely used in jewelry, health care (especially dentistry), and engineering and manufacturing (especially aerospace and automotive) applications for parts with fine features or complex geometry [98]. Traditional molds designed for casting have a dense structure, which makes the cooling stage problematic due to uneven capability in this sense as the casting is wrapped inside a thick sand mold with low thermal conductivity [99]. Also, traditional casting techniques require very high up-front tooling costs together with slow, expensive, and laborious mold manufacturing [100]. In casting production techniques, additive manufacturing has been utilized for the manufacturing of prototypes, patterns (replicas of the final part), sand molds, cores and castings themselves, with an increasing interest in the molds and cores production using AM [99].

Replacing expendable wax patterns with 3D-printed patterns in the process of investment casting (lost-wax casting) can generate substantial cost reduction, even after adding printing equipment and material costs, by significant savings in terms of eliminating labor and materials for injection-molded master patterns, soft inner molds, and wax filling-associated expenses [100]. Literature attests to a wide range of studies for casting materials using 3D printing of inorganic sand molds [101,102,103,104,105], and recent interest is moving towards making the 3D-printed molds out of polymeric materials.

Photopolymerization technologies like SLA produce smooth and ultra-fine structure detailed parts and are consequently a compatible technology to manufacture smooth and detailed molds [100]. SLA materials are available as casting resins containing wax for direct investment (lost-wax) casting, which can be “burnt out” at the end of the process, ensuring intact molds. Formlabs offers a solution in this sense as well, through their low or high wax content resins for casting miniature parts design from ultra-fine structures (i.e., wire filigree) to wide range (stone holes or engravings) [106]. Long before the 3D printing era, vulcanized rubber molds were a major advancement in serial production, allowing investment casting at scale [107]. Depending on the requirements in terms of durability, three major rubber classes are available: organic rubber (destined for intense-use wax molds as it has the highest tear strength), heat vulcanized silicone rubber (can respond to a high level of detail, but has lower tear strength), and RTV silicon (destined for molding around delicate details, but has the lowest tear strength) [107]. Three-dimensional printing of vulcanized rubber molds for room temperature or even high temperature can be used for the production of wax models’ quantity for investment casting wax of miniature-size metal parts [108].

Recently, Fraunhofer IPA researchers combined additive manufacturing and injection molding to create the Additive Freeform Casting process which benefits from the advantages of both technologies. They utilized the FDM process to print a mold (shell) using water-soluble polymer, polyvinyl acetate (PVAc), which was afterwards filled with polyurethane or epoxy resin and dried or cured, respectively. The shell was removed by water immersion [109,110]. This combined free-form casting was found to bring advantages when large, complex components are required in small quantities, while also saving weight.

Although casting using hard traditional molds ensures replication accuracy to the nanometric level, these methods have a major disadvantage when complex designs are needed, as they require the use of multipart or articulated molds and demolding becomes challenging [111]. Koivikko and Sariola [111] tested different sacrificial molds made of dissolvable materials (HIPS, ABS, polyvinyl butyral-PVB, PVA) to cast silicone elastomers. The 3D-printed molds fabricated by fused filament were subjected to dissolution in limonene, acetone, isopropanol/ethanol and water, applying different magnetic stirring and ultrasonication methods in order to evaluate their effect on dissolution time. ABS, PVB, and especially PVA exhibited successful behavior; however, PVA-water is the material-solvent team that is based on non-hazardous components and exhibits suitable dissolution rates, with no secondary effect on the casted elastomer (although HIPS exhibited the fasted dissolution time, limonene caused swelling and cracking in the elastomer during the drying stage). The proposed solutions allowed the manufacturing of overhangs and channels via single-step cast.

Polyvinyl alcohol, derived from the hydrolysis of polyvinyl acetate, is also one of the most accessible polymers from a technological and economical point of view for mold development in both business and the Do-It-Yourself sector. Three-dimensional-printed PVA molds allow the casting of highly detailed objects from metal fluid (mix of metal grit in a resin binder that resembles bronze-like metals perfectly) that could not be made with any other DIY or low-cost casting method, as it is incomparable, easier, and time-efficient compared to using mold making and metal casting [112]. Designer Eliza Wrobel made disposable 3D-printed PVA molds to cast a highly detailed figurine. The PVA molds, printed using a ZMorph 2.0 SX multitool 3D printer, have the advantage of being ready-to-use, not deforming once the material starts to give back heat, and dissolving completely after 24 h water immersion. The cast figurines only needed sanding to remove resin residues and 3D printing layers [112].

Polymer 3D printing of molds extended its use even in the more sensitive fields, like medical implants. In 2016, in a preclinical study conducted in Singapore, Tan et al. [113] obtained excellent cosmetic and cranial models results with patient-specific polymethylmethacrylate PMMA implants produced with low-cost 3D-printed PLA molds. In 2017, the subject was applied in a clinic study, when a team of medical doctors at Joseph University of Beirut [114] adopted a similar route by printing single-piece molds from low-cost PLA and using them to cast a customized PMMA cranioplasty implant, the applied work concluding that the technique is a cost-effective one for delayed reconstruction of various cranial defects. Three-dimensional prints of anatomical structures could be produced with sub-millimeter accuracy (<0.5 mm) compared to the original specimens. Although the low-cost desktop printers for PLA can facilitate the access to this rapid prototyping technology, the major disadvantage of applying this technique in medical fields and hospitals is the need to master software programs by which the digital model of the mold is designed. However, this drawback seems to become less and less major; considering the high demand for 3D-printed tooling, the programs are constantly improving into more user-friendly versions.

Still in the medical field, but towards pharmaceutical applications, Ajmal et al. [115] cast tablets of indomethacin in hydroxypropyl methylcellulose (HPMC) and polyethylene glycol (PEG) formulation using commercial PLA molds 3D printed via FDM with four different designs (with a designed disintegration functionality, composed mainly of two parts: a detachable cylinder and base/lid, which would separate into up to six sections) established by CAD software. The PLA molds’ surfaces were lubricated with corn starch for easier tablet removal; after drying the tablets at room temperature for 24 h, the 3D-printed cylinder parts were removed and the tablets were detached from the molds (base/lid part) using a scalpel. The experiments showed that the resolution influences the ease of detachment in this method and proved in laboratory scale that fast customization of patient-oriented pharmaceutical products can be successfully achieved by means of rapid prototyping.

3D printing allows complex geometry and tailoring of different properties for the optimization of the casting process especially in terms of easier and damage-free demolding. Lv et al. [116] experimented with an innovative damage-free demolding method using a soft ultra-fine mold made of polycaprolactone deposited via electrohydrodynamic printing on a substrate in a predesigned printing path with high precision, used for the effective casting of bio-hydrogels and tested for potential applications in microfluids and cell patterns. The soft ultra-fine mold was framed and hydrogel precursor was poured into the frame and cured. After the mold was detached from the substrate, the fibers were softly peeled from the hydrogel with almost zero damage. The method allowed the damage-free detachment of the generally brittle bio-hydrogels by reducing the demolding stress, with the method showing potential to evolve as a general technique for micro/nanofabrication of brittle materials.

### 4.3. 3D Printing of Molds for Thermoforming

Thermoforming is a widely used technique in the processing of thermoplastics (generally ABS, PET, PETG, HIPS, PC, PP, PE) that involves heating of a plastic sheet over a specific design tool (mold) so that it takes the design of the tool, which is intensively utilized in packaging and consumer goods products, but also extended to automotive, transport or other high-tech industries.

For the manufacturing of parts needed in small quantities, tools made of hardwood are generally used and exhibit satisfactory behavior, while higher quantities, which implicate superior wear stresses, and metallic materials, such as aluminum, are used for tools. Traditional molds require additional processes such as drilling and milling, performed with the use of robust equipment with high investment; therefore, the process can become cost-effective when mass production of parts is needed. Small quantities require the use of molds that are easy, quick, and inexpensive to manufacture. Therefore, additive manufacturing seems to be the perfect solution in this sense as well.

Thermoforming can be performed using vacuum pressure (ideal to obtain parts precisely formed on one side), around 6.9 bar (for complex and intricate details, with surface finish similar to injection molding), and mechanical forming (negative and positive molds are pressed together, ideal for deep profiles). Thermoforming is mostly used for thermoplastics. Once again, Formlabs developed their own guidelines and cases for the optimization of thermoforming via 3D printing of molds or tooling, made of PS, PC, ABS, and HIPS, PETG, PE, and PP, which were evaluated for the replacement of aluminum molds for low-volume manufacturing [117].

When designing a thermoforming tool for 3D printing (Figure 7), both the principles of thermoforming and the ones of additive manufacturing should be considered. Three-dimensional-printed molds can ensure the same features as metal molds, but allow increased design freedom with more intricate geometries [117]. Thermoforming tooling requirements are related to their successful resistance to assembly, forming, and demolding forces, temperatures, any coolants or mold release agents. Depending on the number of parts to be thermoformed, the design, and the product requirements, the Formlabs resin used to build the 3D-printed mold choice can be draft resin—for a quick simple design iteration of large parts and one or more pieces, lower resolution but up to four times faster than standard materials; grey resin—for high surface finish quality and detail parts in one or more pieces, better accuracy, consistency, simpler support removal; rigid 10 K resin—industrial-grade, highly glass-filled material capable of forming limited series of dozens of parts with close to production cycle times, high HDT values (up to 218 °C), and tensile modulus (10 GPa), it is suitable when conditions of forming are challenging [117]. Formlabs tested thermoforming of thick PS sheets for up to 50 cycles, using 3D-printed molds from Rigid 10 K Resin with cooling channels embedded, with execution times shorter by 3–7 times and costs reduced in half compared to traditional tooling, which exhibited quality similar to aluminum tooling [117]. For materials with stronger performance, consisting of PC, Formlabs tested molds 3D printed from draft resin and grey resin, exhibiting a production time of 1 day and production cost lower than USD 400. For the testing of ABS and HIPS molds, Grey Resin, Rigid 10 K Resin, and High-Temp Resin at 100 microns layer height were used, achieving quality similar to those achieved with traditional tooling. For the PETG, P,E and PP thermoforming, up to 20 parts of each were manufactured from Rigid 10 K and Grey Resin molds, without reaching mold degradation. For thinner sheets, after around 10 iterations of short cycle time, demolding issues started to appear, while with ticker sheets produced using longer cycle times, there were no demolding issues and quality was superior [117].

Chimento et al. [118] have used 3D-printed molds manufactured from Zcorp 3DP Zp130 (mixture of plaster, vinyl polymer and sulphate salt [119]) that were subjected to post-processing using diluted cyanoacrylate (CA) and steam to increase strength while maintaining a porous surface suitable for thermoforming, and Zcorp 3DP Zp140 designed for water curing. The Zcorp-printed parts with different post-processing treatments were compared to the traditional mold material—plaster of Paris (calcium sulfate hemihydrate). Zp130 CA treated shower flexural strength comparable with 100% plaster samples, while exhibiting smaller wear areas. In addition, no differences in thermoforming performance were observed between the rapid prototyped specimen and traditional plaster specimens. All the results indicate that 3D-printed molds are feasible for thermoforming prosthetic and orthotic devices such as prosthetic sockets while providing new flexibility, confirming once again that high customizability prosthetic/orthotic devices can be easily fabricated by 3D-printed materials for rapid tooling.

Junk et al. [120] tested rigid PVC and PS sheets for thermoforming over an automotive shape mold produced via 3D printing, concluding that although materials-associated costs were higher than conventional aluminum or hardwood molds, the manufacturing process hourly rate decreased to 19% and process overall costs decreased to 14% of the metal mold-based process values. Besides economic considerations, the design can be easily modified, by adding channels, holes or other additional geometry (spacers), and additional operations related to the mold post-processing (such as drilling, CNC preparations) are completely removed.

Serrano-Mira et al. [121] analyzed the feasibility of using low-cost AM techniques as rapid tooling techniques to obtain thermoforming molds to quickly manufacture small production batches of tactile graphics. They compared two low-cost AM techniques, 3DP with cyanoacrylate infiltration and FDM with PLA, analyzing geometrical reproduction of the molds and their suitability for 0.2 mm thick PVC sheets thermoforming of tactile graphics. When printing small batches (tens of parts), 3DP appeared to be fast (approximately four times faster than PU prototypes) and economical, while FDM with low-cost equipment appeared to be slower, but implicated lower materials and operating costs. Also, compared to 3DP, FDM offers decreased results regarding details reproduction required in tactile graphics, although this issue can be improved by using smaller diameter nozzles and tailoring parameters.

Literature attests to a multitude of both research studies and small-scale production cases in which thermoforming and vacuum-forming methods are performed with the use of 3D-printed tools; this review points out some of the most diverse found.

### 4.4. 3D Printing of Molds for Composites Fabrication

Traditional manufacturing methods for fiber-reinforced polymer matrix composites (FRP) require hard tooling for the molds or mandrels shaping the obtained part. In the thermoset polymers area, one of the most important fields that uses molds extensively is the composite fabrication, which is applied for fiber-reinforced thermoset via vacuum-assisted transfer molding, resin transfer molding, prepreg processing, etc. These techniques can develop at the room temperature and vacuum pressure, or high temperature and supplemental pressure (in autoclave).

Traditional molds for composite fabrication are manufactured from metallic materials (generally aluminum, steel or different alloys) but also non-metallic (specialized tooling materials), but regardless of the raw material they are built from, they require significant labor and machining, and consequently high costs, material waste, and long lead times for even relatively simple part shapes. In this case, FDM printing demonstrated that it could ensure significant cost and time reduction, while allowing design flexibility as well as rapid and easy iteration even when complex geometries are required [122].

For the production of composite materials, different mold architectures are implemented to obtain different types of geometry of the composite parts [123]:one-part mold—used in vacuum bagging methods (i.e., for hand lay-up, resin infusion, prepregs, etc.) and generally for parts that need a glossy finish for one of the sides;two-parts mold—used in compression molding for parts that need both sides with a glossy finish;bladder mold—used in pressure molding where one side is the mold, the other is the bladder surface, for complex geometry that cannot be achieved via vacuum bagging or compression molding due to the impossibility of demolding the composite;mold pattern for negative mold—used when multiple molds are needed for production increase, a single pattern can be used to manufacture several molds.

Formlabs mentions some major factors to be considered in terms of the designing of the molds for composite fabrication such as draft angle, minimum radius, the inclusion of locating pins and indents, inclusion of surface overrun, adding trim lines, all intended to ease the process of technological challenges (i.e., ease of demolding, precise alignment, air entrapment avoidance, repeatable quality, etc.). After design fractures are established, there are also technology-related factors that need to be considered, such as the use of the smallest layer height to optimize the resolution and demolding, the use of a release agent for ease of demolding, avoiding the use of supports on molding faces not to interfere with surface finish, and allowing resin to degas to avoid air inclusion [123]. Formlabs presented three case studies using 3D-printed molds for composite fabrication [123]. The first one was the development of three-layered carbon fabric epoxy composite by hand lay-up and vacuum bagging using their Tough 1500 Resin to 3D print the mold via the SLA process; in the end, compared with outsourced CNC-machined molds, the 3D-printed mold took 2 days to be produced compared to 4–6 weeks. With CNC machined, the total cost of mold production was 310 USD compared to 900 USD for CNC machined. The second one was the development of bidimensional carbon fiber-reinforced epoxy composites prepregs in autoclave using their High Temp Resin on an SLA printer, estimating that the mold would withstand around 10–15 similar cycles, due to the high temperature and pressure in the autoclave. Although it is certainly not suitable for high-volume production, it can be used for dedicated high-performance applications such as dedicated sports equipment, customized tooling for aerospace or personalized prosthetics. The third case analyzed the 3D printing of patterns to cast molds for large series productions of prepreg composites, using their High-Temp Resin with an SLA printer. Comparing the costs to CNC machining, the 3D printing labor time extended over 1.5 h at a cost of USD 300 compared to 5.5 h at a cost of USD 1100 for CNC, mold materials cost USD 50 for 3D printing compared to USD 220 for CNC, while the total cost of the process cost USD 350 for printing compared to USD 1320 for CNC. The costs were reduced around four times on a basic part when using printed pattern for molds.

Stratasys successfully applied its FDM technology for tooling applications to manufacture and repair different composite lay-up configuration in low-volume quantity. However, the materials limitation delayed the progress of this application as the prepreg required temperature in the autoclave exceeding 180 °C was widely used in aircraft structures. Until they developed ULTEM 1010 resin, based on high-performance polyetherimide, able to withstand temperatures above 200 °C without deformation under mechanical loads [124], Stratasys offered ABS, PC, and ULTEM 9085 materials as alternatives for withstanding temperature values up to 85 °C, 135 °C, and 150 °C, respectively. Although PC and ULTEM 9035 HDT cover the 120–125 °C curing temperatures required by CFRP in the autoclave, the use of ULTEM 1010 manages to successfully minimize thermal expansion impact [122]. The guidelines of FDM-printed ULTEM 1010 tooling to build CFRP offered by Stratasys took into account some key considerations for design, material, and testing of the tooling characteristics. ULTEM 1010 performed successfully under harsher flexural loading conditions (using a lower threshold for acceptance) for the equivalent of dozens of high-temperature and -pressure autoclave cycles, anticipating they could exceed 100 cycles with successful behavior of the tooling, the use of lower pressure, and temperature conditions increasing the number of cycles even more [122]. In addition, the data presented suggest that for the vacuum bagging only small pressures (out-of-autoclave) method widely used in aerospace parts production, tool life ceases to be a major problem from creep-induced deformation perspectives [122].

Besides tool life and thermo-mechanical performance on several cycles, when composite materials are manufactured in molds, the materials compatibility is very important, so that debonding of the part from the mold does not generate issues. There are several polymer alternatives to be analyzed for 3D printing the molds for this application. For example, polyethylene terephthalate glycol (PET-G) is highly recommended due to its good capability to detach from the epoxy resin, while ABS molds should be avoided as detachment of the epoxy resin composite could be problematic [125].

In 2016, Oak Ridge National Laboratory collaborated with a team of industry partners to 3D print and machine several large molds and test them in Boeing’s industrial autoclaves to produce carbon fiber composite. The thermoplastic molds survived the high-temperature, high-pressure conditions in the autoclave, which is used to cure aerospace-grade composite parts [126]. The successful testing resulted in high-quality composite parts that can be used in primary aircraft structures. Furthermore, the tools can be re-used to produce part replicates—resulting in further time and energy savings [127]. Different tools made from (PPS) with 50% by weight carbon fiber and Polyphenylsulfone (PPSU) with 25% carbon fiber were developed. The initial tests performed on the molds intended to stabilize the polymer system to withstand variable exposure to elevated temperature without substantial changes in the polymer viscosity. The printed tools were used to fabricate aerospace-grade epoxy reinforced by eight layers of carbon fiber. The tools were cleaned using isopropyl alcohol (IPA), and then three coats of mold release (Frekote 700NC) were applied to the mold surface. The pre-impregnated epoxy/carbon fibers were layed-up in the molds and vacuum bagged. The tools were placed in a production autoclave and exposed to a two-hour cure cycle of 176.6 °C and 620 kPa (90 psi). The tools were scanned after the autoclave process and dimensional analysis and deviation measurements were performed showing that deformations did not exceed 0.1 mm at the composite layup area. The project demonstrated the viability of using additively manufactured parts in the tooling industry to significantly reduce manufacturing costs and energy requirements by accelerating production times [128].

However, even considering all these advancements, 3D-printed tools are not yet common in serial production of high-temperature, autoclave-cured parts for aerospace, as there is still a need to expand the limited material alternatives and use certified properties and behavior of the tooling for these high-demanding applications. But, significant advancements are steadily developing, as in 2019, CEAD (Netherlands) together with partners produced 17 tools printed with short carbon fiber-reinforced polyethersulfone thermoplastic that have been used for more than two years by GKN Aerospace in Germany, for the serial production of CFRP landing flaps for Airbus (France) A350 aircraft, moving even further in 2023 by producing the tools using advanced tape layer additive manufacturing, that involve long fiber in tape form instead of previously used short carbon fibers [129].

On a more research- and education-oriented level, Dynamism, leading provider of professional 3D printing and Industry 4.0 solutions for enterprise, industrial, and education applications, describes the development of the bare-bones carbon fiber process without the specialized equipment needed for more technical processes and high-temperature epoxies. They recommend some major guiding steps: the mold needs to be prepared with a release agent (i.e., PVA helps to smooth out layer lines while providing a reliable release from the epoxies), as it allows its use with most conventional resin systems (epoxy, polyester, vinylester); this method is well compatible with hand layup with or without a vacuum bag, in case resin infusion is required; considering that 3D prints are not 100% airtight, the use of an envelope bagging method needs to be considered; if prepregs are used, the high temperature required for curing makes the PRT-G mold incompatible with the process, as the stress of the vacuum bag will lead to excessive warping and distortion [125].

Besides the widely known application in aerospace parts, CFRP can be used in other various fields (i.e., medical domain). Munoz-Guijosa et al. [130] presented their study on rapid printing of molds for lamination and autoclave curing of epoxy/carbon fibers composite based in prepregs for customized articular orthoses. The molds were manufactured via fused deposition modeling from PLA. In order to respond to the requirements of the epoxy prepregs curing and lamination, in accordance with final product properties related to ankle immobilizing, supporting, or protecting splint, the molds need to meet strict geometrical, mechanical, and thermal specifications. Therefore, the molds need to withstand the mechanical loads generated by the contraction of laminate during the curing process, and those related added by pressure during temperature curing (0.1–0.8 MPa in vacuum bagging or autoclave) maintain the required stiffness and strength at the curing temperature (that may be above 180 °C when an autoclave is used), and exhibit small mean surface roughness (0.5 µm order) to ensure ergonomic/esthetical properties. Considering these requirements, the design must compensate for the drawbacks related to rapid prototyping of molds. The authors created ABS outer shells of the mold, using a precise dual-extruder BCN3D Sigma machine, and in the case of vacuum/pressure curing, the shells were designed with flat areas for the attachment of the supplementary materials needed (release films, breathers, vacuum valves, sealing tape). After printing of the shells, the lamination surface is coated with epoxy resin to tailor the surface roughness. The printed mold shells are filled with a plaster slurry (hardened and dehydrated at 50 °C/2 h), having the role to improve mechanical endurance and heat absorption capacity. Therefore, the thickness of the ABS shell must be minimum (1 mm in this case) as it gives the desired geometry, but the thermal and mechanical properties are ensured by the plaster core. Compared with mold manufactured by machining aluminum, the proposed rapid tooling process ensures almost the same roughness (0.5 µm compared to 0.4 µm for Al), and costs reduced more than 30 times. Although the 3D-printed molds are estimated to withstand 5–10 cycles, considering the customization for each personalized orthoses of patients, the mold is not meant to be used more cycles than the maximum it withstands. The rapid tooling process presented in this paper innovates through the use of conventional FDM of basic thermoplastic polymers ensuring the improved mechanical and thermal properties of the final tooling by filling the 3D-printed shells with clay, making the mold suitable for epoxy CFRP lamination and autoclave curing.

Using 3D-printed molds and patterns in composite fabrication allows businesses to reduce workflow complexity, expand flexibility and design opportunities, and reduce costs and lead time.

### 4.5. 3D Printing of Molds for Tissue Engineering Scaffolds and Medical Applications

Tissue engineering seems to attract extensive research and medicine effort to develop off-the-shelf scaffolds, as they are able to provide a framework for cell proliferation, migration, and attachment, emerging as popular treatments for bone regeneration and wound healing, due to the mechanical properties that support tissue growth, and they also provide a temporary framework for regeneration [131]. Additive manufacturing is a new and emerging field in the tissue engineering sector in medicine. While 3D printing technologies (mainly fusion deposition modeling, stereolithography, laser sintering, inkjet printing) have been clinically deployed in cranio-maxillo-facial surgery, they are primarily used in the areas of models, guides, splints, and implants [131,132]. The three main approaches to 3DP in tissue engineering are bioprinting (printing live cells), printing acellular scaffolds, and printing molds to be filled with engineered tissue [131,133]. One of the first preliminary studies regarding the use of molds manufactured by 3D printing for scaffolds fabrication for bone regeneration [134] used cryogel together with 3D printing to create CT-derived, patient-tailored molds for scaffold fabrication. However, without sacrificial molds, the debonding resulted in the scaffold damage when the mold was opened. Therefore, the group of researchers advanced towards sacrificial (dissolvable) 3D-printed molds, manufactured from PVA, ABS, and HIPS, which dissolve in water, acetone, and d-limonene, to be used to manufacture tissue engineering scaffolds (cryogels, hydrogels) for cleft-craniofacial defects, which were characterized in terms of porosity, swelling kinetics, mechanical integrity, and cell compatibility. Cryogels were fabricated in PVA and ABS molds, while hydrogels were fabricated in PVA and HIPS molds having 1 mm thickness. HIPS molds required a long time to dissolve (5–8 h), making it difficult to remove the cryogels, being fully formed after 24 h. PVA and ABS dissolved in 2–4 h, but the hydrogels in ABS were very fragile and fractured during removal from the mold. All cryogels maintained accurate shape, and showed spongious morphostructure, mechanical durability with approximately 27 µm average pore size, and 80–87% porosity and good biocompatibility. The nanoporous and brittle structure of hydrogel scaffolds was somehow unsuitable for bone regeneration application, but further improvement studies could mitigate these drawbacks [131].

Sacrificial molds are a very attractive solution for tissue engineering scaffolds; the 3D printing of this type of molds has been studied more and more in the past decade. PVA can be readily printed using FDM printing devices both at the professional and DIY level [135], being intensively used in medical applications due to its cytocompatibility [136,137]. Mohanty et al. [138] studied the 3D printing of PVA via FDM for sacrificial molds to cast elastomeric polydimethylsiloxane (PDMS) polymer scaffolds with structured channels. Printing infill density was tailored between 20 and 80% during the process to obtain different porosities of scaffolds, achieving 81.2% porosity at 80% infill for 150 cm^2^/cm^3^ surface to volume ratio. This was the largest scaffold with so many channels fabricated at that time (75 cm^2^ scaffold with 16,000 interconnected channels. The scaffolds were tested for in vitro hepatocytes cells culture for a 12-day period and the results indicated that the scaffolds produced in 3D-printed PVA molds led to a rapid, cheap, scalable, and compatible with cell culture process. PDMS microfluidic channels structures were fabricated in 3D-printed ABS molds, afterwards dissolved in acetone, resulting in channels down to 90 µm, with 500 µm diameter [139]. In a recent study, Brooks-Richard et al. [140] presented the design and fabrication of MEW (melt electro-writing) tubular scaffolds with complex geometry mimicking patient-specific vascular structures, on FDM 3D-printed PVA molds. The results showed that PVA was a more suitable material than metal mandrel due to its low insulative properties that improve the ability to produce highly ordered scaffolds, which were easy and fast to remove in water without affecting the MEW scaffold fibers’ morphology and alignment.

PVA molds are used in medical applications not only as sacrificial molds. In 2014, a team of medical doctors fabricated an inverse replicate of the normal ear for a template in first-stage microtia surgery. A negative mold of the ear was fabricated using rapid prototyping with PLA, the printing process took 90 min, and required less than 1 USD total cost for disposal, and the mold was sterilized for intraoperative use as a template to create an autologous costochondral implant in its likeness [141].

The stomatology area researched the use of 3D printing of polymeric molds as a tool for their patients’ customized needs. Yang et al. [142] fabricated novel TNZ dental fillers which were indirectly produced by thermal pressing using customized 3D-printed molds, manufactured from commercial filaments of PLA and ABS using a desktop printer.

Three-dimensional printers can produce anatomic models based on 3D ultrasound, magnetic resonance imaging (MRI), and computed tomography (CT) scans [143,144,145]; therefore, they can be successfully used to generate patient-specific molds. MRI investigations are of great use in the development of molds design and CAD architecture, which help by offering predictions for future cases. Pokorni and Tesarik [146] developed molds from PET-G polymer to produce phantoms of the human head tissues (skin, bone, cerebrospinal fluid, brain), to mimic head geometry and evaluate stroke detection mechanisms that can be further applied to patients. The design of the molds was developed from MRI-derived scans. Different shapes and sizes of head forms were 3D printed via FDM with Prusa i3 MK2, using a 0.35 mm layer height and 0% infill for a faster and more material-effective process. The printed molds were hollow, so basanite filler material was used to improve their mechanical strength.

MRI investigation information was used in a medical case presented by Costa et. al. [147], in which the anatomical registration of preoperative MRI and prostate whole-mount obtained with 3D-printed, patient-specific, MRI-derived molds was compared with conventional whole-mount sectioning, the study showing that 3D-printed molds for prostate specimen whole-mount sectioning provides significantly superior anatomical registration of in vivo multiparametric MRI and ex vivo prostate whole-mounts than conventional whole-mount sectioning. The design was composed of several stages, using multiple software (i.e., Matlab for volumetric reconstruction extract and conversion to STL file, Netfabb for molds generation based on a generic, SolidWorks for building of a parametrically controlled three-part slicing mold with holes for fixative perfusion and slots for slicing alignment). In the initial trial, the molds were printed on a commercial-grade ProJet 3510 Plus 3D Systems printer using a UV-curable resin (Visijet Crystals, 3D System), and after the parameters establishment, the MRI-derived molds were fabricated on a consumer-grade 3D printer (Leapfrog Creatr XL) using polylactic acid. Another study of medical cases presented the development of 3D-printed PLA patient-specific molds in a prostate phantom model which reduced the MRI-whole mount registration error relative to conventional sectioning. The 3D-printed molds showed the potential to improve prostate MRI-pathology correlations, with the potential to be applied to other organs [148].

A team of medical doctors from the USA presented an algorithm to automatically create 3D-printed molds guiding medial temporal lobe extraction for postmortem MRI, with interactively positioned cut planes used in four hemispheres, their method reducing errors and dependence on anatomical expertise while allowing more tissue to be spared from each brain donation and enabling postmortem imaging at a larger scale [149].

Still in the field of high-resolution imagining medical tooling, Weadock et al. [150] used 3D-printed molds for shaping bioabsorbable implants for customized surgical orbital repair, improving fit, reducing tissue handling and postoperative edema, and reducing surgical times. The orbital area images captured by computed tomography (CT) techniques were used to create STL models of the molds and were edited to create the mirror of the area and overlap it with the fractured side. Sterile or sterilizable molds printed using Formlabs Form 2 printer were fabricated using the images and taken to the operating rooms and used to shape the customized orbital implant for fracture repair in three patients, using bioabsorbable implants.

Three-dimensional-printed molds from PLA were used to fabricate replicas of uterine and fibroid elements, and a realistic model with silicone material uterus and fibroids was used to help resistant simulated laparoscopic myomectomy at low cost. Previously used molds can be repaired with silicone and reused by other residents [151]. Also, breast reconstructive surgery benefits from the use of 3D-printed molds. Patient-specific 3D-printed templates for intraoperative use were designed based on 3D stereophotogrammetry images. The molds were printed from PLA using an Ultimaker 2 printer and then placed in a sterile plastic sleeve to be used for the fitting of the free flap. Prior to anastomosis, the flap was positioned in this sterile covered template, where the contours of the free flap could be traced with a marker pen along the 3D-printed mold, and sutures can be placed to maintain the flap shape. During breast reconstruction, the autologous flap was placed inside the printed template to aid the surgeon in determining the shape and volume of the autologous flap creating the desired breast dimensions. Patients were 3D-photographed 6 to 9 months post-operatively. The study showed that for both unilateral and bilateral breast reconstructions, a mold can represent a useful, low cost, and fast processing tool added to the autologous reconstruction procedure [152].

### 4.6. 3D Printing f Molds for Soft Lithography

A special use of casting method involving molds is soft lithography, a technique used to create micro devices or three-dimensional structures by means of casting liquid polymer precursor against a topographically patterned mold. Although it involves casting of a polymer, it cannot be included in the general casting molding, as it is not an industrial-type technique, but rather a science-oriented one, as it is broadly used in bio-imprinting and micro/nanofabrication [153]. Soft lithography includes a cluster of methods that uses soft polymeric materials to fabricate small-size devices such as stamps, channels, or membranes with micro-sized features, being a reliable, easy, and low-cost process that allows replicating 3D structures from cm down to micrometric dimensions. The most common devices fabricated with this technique are microfluidics, intensively used in cell biology.

The most common elastomer used in this technique is PDMS, a soft bio-compatible elastomer that has high thermal and chemical stability, low toxicity, chemically inertness, insulating properties, gas permeability, excellent optical transparency to UV and visible light, low cost, mechanically flexible and durable, and last but not least, it is easy to mold [154,155].

Considering its unique properties, PDMS is of great interest in microfluidics applications (widely used in fluid mechanics, reagent mixture, cell biology, particle and cell separation, metabolomics and proteomics, forensic, genetic analysis) as microchips, using soft lithography. However, the costly and time-consuming master mold preparation, the silane surface treatment of the mold required to prevent PDMS detachment problems that can intervene in cell related studies, as well as different required designs of the structure that can be technologically complicated to obtain represent some major impediments. Therefore, 3D-printed molds stood out as an attractive alternative for molds fabrication in soft lithography, methods like stereolithography and digital light processing being some of the most suitable, especially for microfluidics and biomedical areas [156]. Resin or silicone are the generally used materials for PDMS molds fabrication, but beside the fact that they have higher costs than other materials available for 3D printing and require dedicated printers [157], they also generate an effect of inhibition of the curing process of the resin at the contact area of the PDMS with the mold [158], as full curing would be influenced by residual catalysts and monomers [159,160]; therefore, mold surface treatment before PDMS casting remains a challenge even for the 3D-printed ones. Studies attest to the use of different standard pre-treatments of 3D-printed molds via UV curing, ethanol-sonication surface cleaning, preheating, and silanization [156,161] while other studies use alternative treatments such as ink airbrushing [162], a multiple-step procedure including UV treatment, ethanol immersion, air plasma, and perfluorooctyl triethoxysilane treatment [163]. However, the protocols adopted in different research studies seem to be influenced by a variety of factors; therefore, a standard protocol could not be established so far.

Bazaz et al. [156] proposed a method of casting PDMS directly over a 3D-printed mold fabricated directly by the DLP method using a resin based on methacrylated oligomers and monomers, without any pretreatment/surface treatment of the mold, reducing the timeframe for mold fabrication to less than 5 h, compared to several days (for standard soft lithography). Using this resin allowed the removal of mold treatment, as the methacrylated monomers in the resin composition do not react with the casted PDMS, as there are no residual monomer units on the mold surface to impede PDMS polymerization. The PDMS detached from the molds without difficulties. Four microfluidic devices were designed for separation, micro-mixing, concentration gradient generation, and cell culturing applications, the results indicating the biocompatibility of the resin and stable gradient indicating the potential to be used in drug delivery systems.

An Australian research study [164] experimented with a simple fabrication technique of lung-on-a-chip devices using surface-treated DLP 3D-printed molds using photopolymerizable resins based on acrylate polymers for the casting of PDMS parts. The use of acrylate polymer-printed molds allowed a multiple step treatment of their surface (isopropanol washing, UV curing, ethanol immersion, plasma treatment, silanization) in order to prevent PDMS from sticking to the molds and consequently making them suitable for repeated long-term PDMS casting. The approached simple, robust, and cost-effective method allows fabrication of the chip in less than a day, and the use of re-usable molds. In the field of PDMS casting, more advanced studies have developed recently. Yasuda et al. [165] presented the manufacturing of a shark skin-like silicone rubber film that mimics the simplified 2D surface of a shark’s skin. The study developed and optimized 3D-printed molds for silicone rubber casting, choosing a 2D-surface version as first prototype. The 3D printing of the full 3D shape remains challenging as supporters are required for 3D printing overhangs of 30° or smaller relative to the horizontal plane, and these supporters would need to be removed during post-processing. The 3D-printed mold proposed by the authors allows for re-use of the molds to increase the manufacturing output. The mold was printed using PLA 2.82 mm filament on an Ultimaker 3 printer. PDMS silicone was casted into the printed molds, the method enabling production of large surfaces of orientable micropatterned repetitive structures at a very reasonable cost performance.

Once again, one of the alternative materials that belongs to the more accessible class is polylactic acid; besides the cost effectiveness, it exhibits biocompatibility and biodegradability features, which are crucial for the PDMS molds applications. However, for the use in PDMS casting for cellular applications, PLA molds need to be subjected to a further step after printing, for the surface fine details adjustment in order to smooth the rough edges. Van der Borg et al. [166] used 1.75 mm diameter PLA filaments to print molds, using a commercial 3D filament printer for the use in casting of PDMS to study biological samples by light microscopy. Printing parameters used were 190 °C, on a tape-covered metal phase heated at 60 °C, 0.1 mm layer height, 10 mm/s print speed, without supports. After printing, the mold surface and edges were smoothed by heated chloroform vapors treatment and afterwards left suspended in the fume hood for 1 h and placed in a vacuum desiccator for 12 h. PDMS was casted into the assembled molds, desiccated and cured at 60 °C/4 h. After detachment from the mold, PDSM excess was removed with a scalpel, obtaining 3 mm height rings. The results indicated that PLA 3D printing of molds represents a promising alternative to be used as molds for cellular studies. Others developed a modular microfluidic system for PDMS casting in PLA 3D-printed molds for high-resolution imaging and analyses of leukocyte adherence to differentially treated endothelial cultures. The molds for PDMS casting were printed with a Form 2 printer using black resin and layers of 25 µm thickness, and the alignment tool was printed using an Ultimaker 3 Extended printer using black PLA filament in a 0.4 mm nozzle and 150 µm thick layers. PDMS modules casted for microfluidic chips were bonded to glass slides by connection to vacuum. The 3D printing of tools in this study contributed to the optimization of the functionality of modular microfluidic systems, by using customizable, user-designed devices [167]. The soft lithography technique implies a very diverse scientific and technological set-up, being greatly influenced by the specificity of each of the study features (used geometries, materials, and target applications); therefore, it still remains a sector in which trying to identify a generally applicable design and parameters set-up is a challenge.

### 4.7. 3D Printing of Sacrificial Molds

Sacrificial molds are a class of non-reusable molds that can be destroyed after the part has been produced. They can be made of low melting point materials such as wax that are typically destroyed by heating, or by dissolvable materials that can be washed in water or other solvents. Unlike reusable molds for which disassembly and demolding considerations drive the mold decomposition, in the case of sacrificial molds primary considerations for decomposition are manufacturability of individual mold components [168]. When using sacrificial molds, rather than mimic the conventional functionality of a tool, the soluble/meltable tooling uses the same technologies and equipment, but the material that creates the mold is changed. Soluble tooling allows for a flexible workflow from geometry to molds to parts [169]. Besides the sacrificial molds cases already mentioned in the previous section dedicated to injection molding [85,97], casting [111], and tissue engineering [131,136,137,138,140] molding using sacrificial molds does not constitute a stand-alone technology, but it is often used as an alternative to build parts via different customized technological routes, where reusable tools and tool life are not issues to be considered.

Some of the most important motivations when choosing sacrificial molds are encountered in situations like the following:when small size features complex geometries like the ones provided with microchannels or overhangs, seamless or hollow areas are needed;when removing/debonding the part from a fix mold is technologically challenging or generates significant damage to the formed part;when complex geometry requires the use of multipart or articulated molds and demolding becomes challenging;when the volume of production allows the use of molds that become waste once a part is produced.

Sacrificial molds can be used in individual or combined situations as mentioned above. Sacrificial tooling allows designers, engineers, and researchers to create hollow, seamless, and complex structures with smooth internal surfaces and simplified tool removal [170]. Some of the traditional sacrificial molds are made of eutectic salts, ceramics, cast urethanes, or other similar materials, but they are generally difficult to handle due to brittleness, require additional tooling, or are limited in terms of geometries flexibility due to production or removal challenges [170].

Three-dimensional-printed sacrificial molds have been widely used for manufacturing microfluidic channels, polymeric scaffolds, engineering vasculatures, inorganic 3D matrix materials, and microneedles [138,171,172]. In terms of 3D-printed sacrificial molds materials, the alternatives are still even more limited than the ones for reusable 3D-printed tooling; however, the research conducted so far is promising in this sense.

One of the most intensively used polymers for development of sacrificial molds is PVA, as it is a hydrophilic and therefore a water-soluble, biocompatible, mechanically stable with low toxicity compound that can be easily processed as it can be printed at around 180 °C. PVA was often used to produce sacrificial molds for scaffolds made of PDMS [138,171,173], gelatin [174], fibrin or other materials used to produce different small-scale detailed patterns needed in engineering vasculature or other channel networks applications. The fabrication of sacrificial PLA templates or molds is generally performed via FDM printing [175].

Nagarajan et al. [176] presented the use of FDM-printed sacrificial PVA molds to fabricate self-standing water-insoluble gelatin scaffolds with tunable pore size and porosity. Varying the PVA infill density, they obtained porosity values between 400 and 1200 μm, and that proved to be stable in a phosphate-buffered saline swelling agent. Their results show that the sacrificial mold approach allows the fabrication of gelatin scaffolds with tunable pore size and architecture suitable for tissue engineering applications, which could be further extended to customized scaffolds using various other biopolymers or synthetic polymers. Zou et al. [177] used PVA sacrificial molds to fabricate a pre-vascularized face-like construction based on a 3D tai-chi pattern. The PVA mold scaffold was printed by FDM and filled by printing with hydrogel composites (nanocellulose, agarose, and sodium alginate with HUVECs and human fibroblasts), and removed with PBS solution after crosslinking with CaCl_2_. PVA 3D-printed sacrificial templates were also used by Park et al. [178] to produce customized ultrathin tubes with adequate mechanical flexibility to mimic bile ducts. The PVA templates were printed at high temperatures and the surface was smoothed by ultrasonication at 50 °C; they were coated with polycaprolactone (PCL) by immersion and removed by water dissolution and ultrasonication. Another study [179] presented the coating of 3D-printed PVA sacrificial templates, with PCL and TPU for tailored porous surfaces with flexibility compatible with soft tissues. Hu et al. [180] used sacrificial PVA molds printed by FDM for microchannels development in tissue engineering applications, which were embedded into three different matrix materials (matrigel, fibrin, gelatin) and removed afterwards by perfusing.

Like previously mentioned, there are studies that introduced the use of other polymers as sacrificial AM molds, such as ABS, HIPS, and PVB, that can be dissolved in different chemical solvents (acetone, limonene, isopropanol/ethanol), that sometimes generate environmental issues, and might as well affect the produced part if the proper compatibility between mold/part is not taken into consideration [111,131]. Besides these, PLA is another attractive polymer, suitable to be used as sacrificial template. PLA was 3D printed, immersed in a gelatin solution at 4 °C, and dissolved with the use of dichloromethane solvent to form a gelatin template, and gelatin methacrylate solution with cells was used to cast the template, which was subsequently removed as well at 37 °C, resulting in a gelatin methacrylate human tissue model with a microchannel network [181]. Montazerian et al. [182] developed 3D-printed PLA shell molds with superior structural integrity to fabricate porous channel network PDMS scaffolds that were removed in dichloromethane solvent.

Poly(N-isopropylacrylamide) (PNIPAM) is another attractive polymer for biomanufacturing applications, often used in drug delivery, soft robotics and engineered vasculature, due to its biocompatibility, ease of processing, and solubility in water at low temperatures [183]. Lee et al. conducted research studies using thermosensitive PNIPAM as a sacrificial template to fabricate microvascular networks within gelatin scaffolds, removing the mold/template by the solvent-spinning method [184], and further comparing the effects of PNIPAM-fabricated microchannels and macrochannels on the formation of normal functional vessels [185]. PNIPAM sacrificial molds are generally produced by electrospinning with microfibers, increasing the scalability of the 3D-printed sacrificial template [175].

In the same category of thermo-responsive polymers for sacrificial molds, other studies used Poloxamer 407 (also known by the trademark Pluronic F127), a triblock copolymer consisting of a central hydrophobic block of polypropylene glycol flanked by two hydrophilic blocks of polyethylene glycol, a water-soluble polymer, that displays a reversible thermal characteristic, as it is liquid at room temperature exhibiting good printability, liquefies at 4 °C, and takes a gel form when administered at body temperature, which makes them attractive candidates as pharmaceutical drug carriers or complex vascular network sacrificial templates [186,187]. Nothdurfter et al. [188] printed Pluronic F127 as a sacrificial mold, on a layer of crosslinked cell-laden hydrogel and fabricated hollow channels in a micro-jetted cell-laden hydrogel chip, having a PMMA rigid shell to mimic a neuroblastoma tumor-environment model. The Pluronic F127 mold was removed by liquefying below 15 °C. While other studies used liquefication at 4 °C to remove Pluronic F127 sacrificial molds [189,190], some used a Pluronic F127 3D-printed sacrificial mold to fabricate photocurable hydrogel scaffolds with customized channels by printing the photocured matrix and removed the mold by immersion in PBS [191]. Others studied improved the mechanical properties and fidelity of the Pluronic F127 3D-printed mold by adding nanoclays into the composition, followed by encapsulation in PDMS and curing and removal by liquification in water at 4 °C [192]. However, although promising and easy to remove, the weak mechanical properties of Pluronic F127 need to be considered when casting in situ scaffold matrix [193].

Another AM polymer that can be used for sacrificial molds is polycaprolactone, synthetic, semi-crystalline, biodegradable polyester with a melting temperature of 60 °C, which can be dissolved in chloroform, dichloromethane, and dioxane [194]. PCL sacrificial molds were used to produce vascular niches and sweat gland interactive models and were removed by incubation with chloroform after dehydration, leaving behind porous constructs [195]. For sacrificial PCL templates with small-size features, electrospinning and electrohydrodynamic jet printing are often used, being extremely useful for engineering vasculature [175].

Other dedicated polymers can be implemented as sacrificial molds via 3D printing and removing, such as water-soluble Poly(2-cyclopropyl-2-oxazoline) [196], potassium bromide soluble polyelectrolyte complex [197], water-soluble butanediol vinyl alcohol copolymer [198], PDMS [199], petroleum jelly–liquid paraffin [200], and water-soluble thermo-responsive polyisocyanide [201].

In terms of meltable sacrificial molds, wax is one of the most used materials. Three-dimensional microvascular networks within an epoxy polymer matrix were fabricated by casting into 3D printing molds made of sacrificial wax, which were subsequently removed by heating above the melting temperature of 60 °C [202]. However, the melting temperature value of most wax limits the material casted for scaffold formation, as polymers that requires higher curing temperatures than the mold can resist are not an option.

When selecting the material of the sacrificial mold or template for biomedical applications, it is very important to consider the compound and/or temperature for mold removal, as some solvents can damage the material of the scaffold, while high temperatures needed to melt the mold material could exceed the thermal resistance of the scaffold material.

Three-dimensional printing sacrificial templates have shown remarkable potential for fabricating intricate structured engineered vasculatures due to their feasibility and versatility, but there are still studies needed to overcome the challenges in producing biomimetic vasculature, related to building of hierarchical vasculature within tissue engineering scaffolds [175].

## 5. Conclusions

The outbreak of additive manufacturing use in almost all industries worldwide changes not only the form, functionality, and pathway of products to the market, but also the methods and routes that are followed to build the products, revolutionizing the way the products are created. Additive manufacturing application in mold development ensures an important mitigation of the area in which traditional manufacturing exhibits limitations, allowing the development of unique designs and tools that can be continuously customized and adapted according to the application requirements and customer’s needs while maintaining the reduced costs and time provided through its characteristics. Three-dimensional-printed molds bring significant benefits in industrial and business fields, as they contribute to the optimization of supply chains and business strategies in small- to medium-scale production in industries like automotive, aerospace and transport, electronics and construction. In the special use applications from tissue engineering and biomedicine, the use of 3D-printed molds allows high quality and detailed customization of dedicated or individual-use products that would not be achievable by traditional techniques. From complex geometries to mass customization, 3D-printed molds can provide significant technical and financial advantages for the manufacturing process and quality of obtained products. Three-dimensional printing of molds is encountered in laboratory research studies, small- to even large-size (in some situations that allow it) industries as well as companies that developed offering 3D printing services for other beneficiaries. The increasing availability of 3D printing services allows researchers without expertise in design or manufacturing to acquire molds already customized to their required characteristics and produce their own devices at low cost, while experienced researchers in the field can fabricate and customize the molds and continuously adapt them for their specific applications.

## Figures and Tables

**Figure 1 polymers-16-01055-f001:**
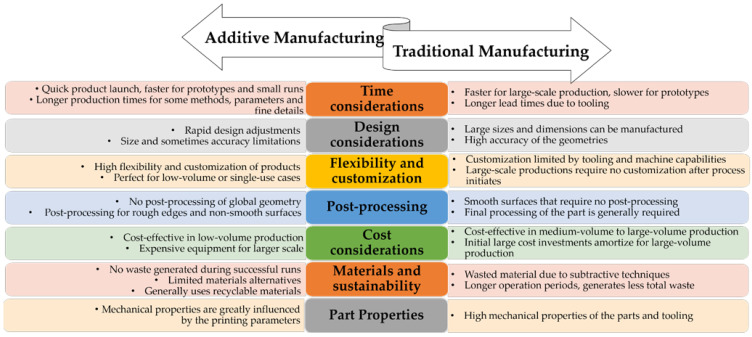
Main advantages versus main disadvantages of additive manufacturing compared to traditional manufacturing techniques.

**Figure 2 polymers-16-01055-f002:**
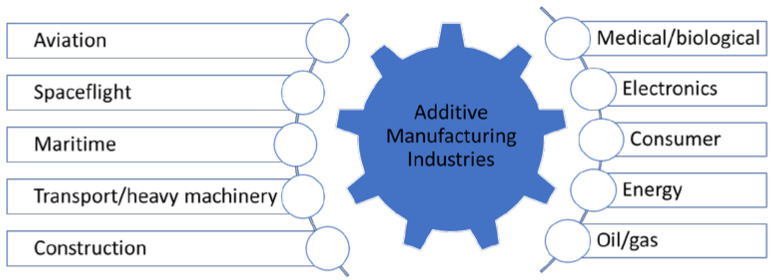
Main industry domains in which additive manufacturing is used.

**Figure 3 polymers-16-01055-f003:**
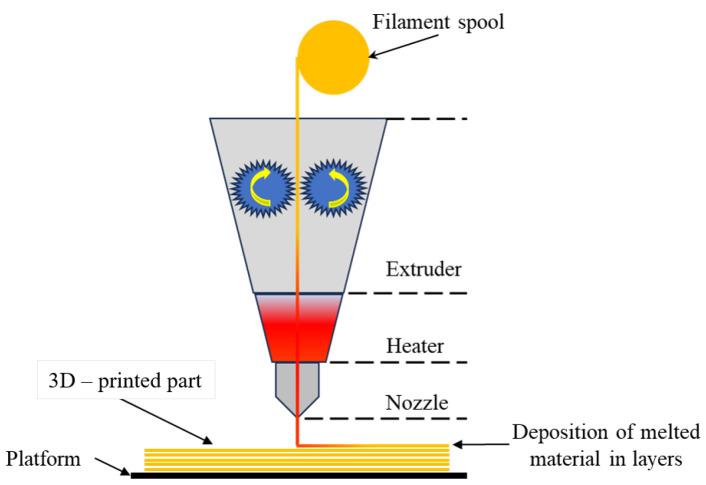
Working principle of the FFF/FDM technique.

**Figure 4 polymers-16-01055-f004:**
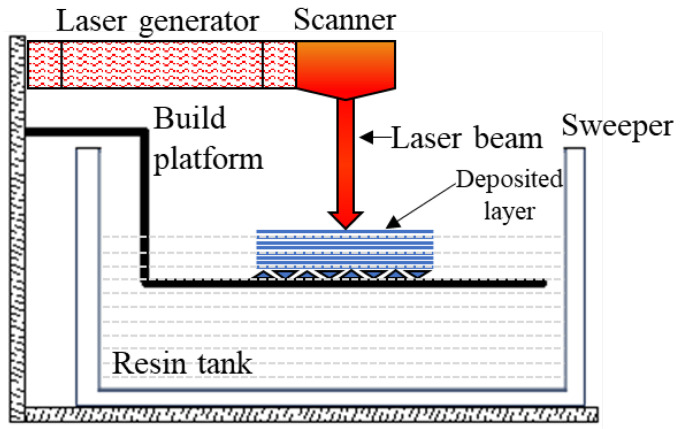
Working principle of the SLA technique.

**Figure 5 polymers-16-01055-f005:**
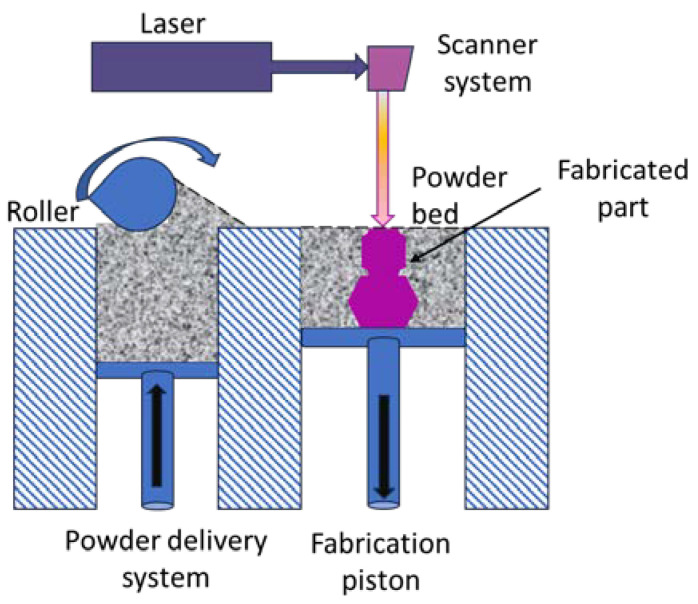
Working principle of the SLS technique.

**Figure 6 polymers-16-01055-f006:**
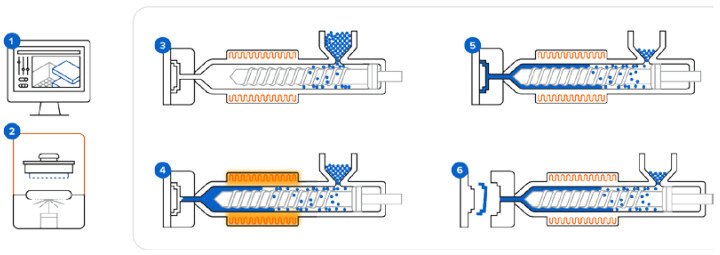
Workflow of injection molding process with 3D-printed molds: 1—mold design; 2—mold 3D printing; 3—mold clamping; 4—injection; 5—cooling; 6—demolding (image reproduced with Formlabs’ permission [73]).

**Figure 7 polymers-16-01055-f007:**
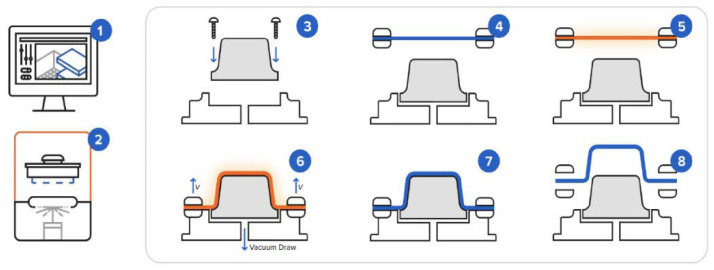
Thermoforming with 3D-printed molds (1—Mold design, 2—Mold 3D printing, 3—Mold assembly, 4—Sheet clamping, 5—Heating, 6—Forming, 7—Cooling, 8—Demolding and trimming) (image reproduced with Formlabs permission [117]).

**Table 2 polymers-16-01055-t002:** Main polymers used in FFF/FDM 3D printing—advantages, disadvantages, applications [20,26,27,28,29,30,31,32,33,34,35,36,37].

FFF Thermoplastics	Advantages	Disadvantages	Applications
PLA	Biodegradable, easy to print, cost-effective	Low strength, low durability, brittle	Consumer goods, toys, DYI, packaging, biomedical
ABS	More durable than PLA, impact-, heat-, chemical-, abrasion-resistant	More challenging to print, prone to warping	Consumer goods, tools, automotive, electrical enclosures
Polyamides	Durable, high strength, flexible	Water uptake, delamination, and poor adhesion when filled	Prosthetics, tools, encapsulations, working prototypes, mechanical components
PET-G	Versatile, flexible, mechanical strength, easy to print	Prone to dampness, easily scratched	Packaging, mechanical parts, printer parts, protective components
TPU	Rubber-like, flexible, durable	Challenging to print	Seals, gaskets, automotive, medical supplies
HIPS	Strength, flexible	Only compatible with ABS, easy to recycle, good support material	Protective packaging, containers
PVA	Biodegradable, cost-effective	Moisture uptake	Support in overhanging parts, sacrificial molds
PPS	Mechanical strength, thermally stable, chemically resistant	Low T_g,_ brittleness, low impact strength, prone to warping without fillers	Mechanical parts
PEI	High T_g_, flame retardant, mechanical strength	Expensive, susceptible to cracking	Automotive, aircraft parts
PEI/PC	High T_g_, thermally stable, mechanical strength, chemically resistant	Water uptake	Transport, automotive, space applications
Carbon, glass, aramid fibers composites	Rigid, strong, tough	Compatibility limited to expensive industrial FDM 3D printers	Functional prototypes, jigs, fixtures, tooling

Where: HIPS—high-impact polystyrene, PVA—polyvinyl alcohol.

**Table 3 polymers-16-01055-t003:** Main polymers used in SLA 3D printing—advantages, disadvantages, applications [20,44,45].

SLA Resins	Advantages	Disadvantages	Applications
Standard	High tensile strength, high resolution, smooth surface finish	Very brittle (low elongation at break)	Visual prototypes, art models, concept models, looks-like prototypes
Tough (ABS-like)	High stiffness, excellent resistance to cyclic loads, compromise between properties of durable and standard resin	Not for parts with thin walls (minimum 1 mm), low HDT, brittle (low elongation at break)	Functional prototypes, mechanical assemblies, rigid parts that require high stiffness, housings and enclosures, jigs and fixtures, connectors, wear-and-tear prototypes
Durable (PP-like)	Highest impact strength and elongation at break, wear-resistant, flexible	Not for parts with thin walls (minimum 1 mm), low HDT, low tensile strength (lower than tough resin)	Prototyping parts with moving elements and snap-fits, consumer products, and low-friction and low-wear mechanical parts, housings and enclosures, jigs and fixtures, connectors, wear-and-tear prototypes
Heat-resistant	HDT between 200–300 °C, smooth surface finish	Poor impact strength, brittle, not for parts with thin walls (minimum 1 mm), temperature resistance increase decreases elongation	Heat-resistant fixtures, mold prototypes, hot air, gas and fluid flow equipment, and casting and thermoforming tooling, heat-resistant mounts, housings, and fixtures, molds and inserts
Ceramic-filled	Very stiff and rigid (high modulus and low creep), very smooth surface finish, good thermal stability and heat resistance)	More brittle than the tough and durable resins, brittle (low elongation at break), low impact strength	Molds and tooling, jigs, manifolds, fixtures, electrical application housings, and automotive parts
Flexible and elastic resin (rubber, TPU, silicone-like)	High flexibility (high elongation at break), low hardness (simulates an 80A durometer rubber), high impact resistance, flexibility of rubber, TPU, or silicone, bending, flexing, and compression resistance, repeated cycles without tearing	Lack the properties of true rubber, require extensive support structures, UV radiation sensibility, not for parts with thin walls (minimum 1 mm)	Objects that will be bent or compressed, wearables prototyping, multi-material assemblies, handles, grips, and overmolding, consumer goods prototyping, compliant features for robotics, medical devices, and anatomical models, special effects props and models
Clear resin	Polishes to near optical transparency, moisture-resistant, durable, large format available, stiff	Requires secondary operations for functional part clarity	Parts requiring optical transparency, millifluidics

**Table 4 polymers-16-01055-t004:** Main polymers used in SLS 3D printing—advantages, disadvantages, applications [20,47,48,49].

SLS Resins	Advantages	Disadvantages	Applications
PA12	Strong, stiff, durable, impact-resistant and can endure repeated wear and tear; Resistant to UV, light, heat, moisture, solvents, temperature, and water	High porosity and low molecular weight deteriorate its mechanical properties, especially ductility and toughness	Functional and high-performance prototyping, end-use parts, medical devices, permanent jigs, fixtures, and tooling
PA11	Similar to PA12, but higher elasticity, elongation at break, and impact resistance	Lower stiffness than PA12	Functional prototyping, structural end-use parts, jigs, and fixtures, snaps, clips, and hinges, orthotics and prosthetics
Glass-filled PA12	Enhanced stiffness and thermal stability	More brittle, reduced impact resistance and flexibility	Robust jigs, fixtures, replacement parts, parts subjected to sustained loadings and high temperature, threads, and sockets
Carbon fiber-filled PA11	Highly stable, lightweight, high-performance material	More brittle, reduced impact resistance	Replacement for metal parts, tooling, jigs, fixtures, high-impact equipment, functional composite prototypes
Mineral-filled PA	Enhanced thermal properties, dimensional stability, rigidity, high HDT	Reduced impact resistance and flexibility, rougher surface than unfilled PA	Parts to withstand high temperatures or mechanical loads
Aluminum-filled PA	Dense, thermal, and conductive properties	Reduced impact resistance and flexibility	Parts with a metallic appearance, mechanical parts that do not experience high stress
Polypropylene	Ductile, durable, chemically resistant, watertight, weldable	Not as strong or rigid as other 3D-printed materials	Functional prototyping, end-use parts, watertight housings, cases, packaging prototypes, medical devices (orthotics and prosthetics), automotive interior components, strong and chemically resistant fixtures, tools, jigs
TPU	Flexible, elastic, rubbery, resilient to deformation, high UV stability, great shock absorption	Limited heat resistance, moisture sensitivity	Functional prototyping, flexible, rubber-like end-use parts, wearables and soft-touch elements, padding, dampers, cushions, grippers, gaskets, seals, masks, belts, plugs, tubes, medical devices (soles, splints, orthotics, prosthetics)
TPE	Elasticity, resistance to abrasion and good UV and ozone resistance	Temperature-sensitive, prone to shrinking	Seals, gaskets, plugs, grips, handles, over-molds, tubes, masks, and gloves
PEEK, PEKK	Excellent mechanical strength, stiffness, chemical resistance, wear resistance, thermal resistance	Low resistance to UV light, low flexibility, expensive	Components subject to friction or wear, surgical tools and implant, applications that require superior thermal resistance

**Table 5 polymers-16-01055-t005:** Methods of producing polymer prototypes using different tooling methods [70,71,72].

Prototype Production Methods	Mold Durability	Average Mold Cost	Average Cost/Part	Production Average Cost	Lead Time	Design Flexibility
FDM direct 3D printing	N/A	N/A	Low to high	Low to high	Short to medium	High
Conventional Molds and Tooling	High (>10,000 parts)	High (2000 USD)	Low	High	Long	Low
3D-Printed Polymer Molds and Tooling	Low (1–10 parts)	Low (50–80 USD)	Low to medium	Low	Short	High
3D-Printed Metal Molds and Tooling	High (>10,000 parts)	High	Medium to high	Low to High	Short to long	Low

## Data Availability

Data are contained within the article.

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
