# Peer review of "The Use of Additive Manufacturing Techniques in the Development of Polymeric Molds: A Review"

_polymers, 2024, doi:10.3390/polym16081055_

Round 1
Reviewer 1 Report
Comments and Suggestions for Authors
In this paper, the authors attempt to review the available studies related to 3D-printed polymeric molds and evaluate their properties and applications. Considering the increasing usage of 3D-printed molds, this subject is interesting and I’m sure it will be insightful for the journal’s readers. However, there are some points that the authors need to address before suggesting it for publication.
1- In Figure 1, the authors discuss the advantages and disadvantages of additive manufacturing (AM). They mention fast production time and cost-effectiveness as advantages of AM compared to conventional methods. However, in general, this is not true, especially when compared to other methods such as injection molding, as AM often takes more time, especially in moderate and large-scale production. The same can be claimed for cost. Additionally, one of the main disadvantages of AM is that 3D-printed parts may lack sufficient mechanical properties. Please consider revising this figure.
2- To enhance the quality of the information presented in Table 1, please consider adding the wire arc and DED (Directed Energy Deposition) methods as important techniques for metal additive manufacturing (AM).
3- Some of the abbreviations are defined multiple times. Generally, they need to be defined only at the first occurrence. For example, FDM is defined in Table 1 and also in line 84. FFF is defined in Table 1, and also in lines 85, 86, and 90. Please review all instances of these abbreviations in the paper and correct this issue.
4- It would be beneficial if the authors also discussed the main requirements of materials used for 3D printing molds. Additionally, they should consider discussing the considerations that need to be satisfied in the design of 3D-printed polymeric molds.
5- One of the main applications of molds is in composite fabrication. Despite mentioning applications for 3D printed molds, the authors didn’t consider this significant application. I suggest including the application of 3D-printed polymeric molds in composite fabrication. For reference, ULTEM 9085 is commonly used for composite tooling. Additionally, Stratasys is one of the pioneers in the application of 3D-printed molds for composite manufacturing.
6- In recent years, some companies have utilized 3D printing to produce sacrificial polymeric molds or smart molds for fabricating structures with complex shapes. I recommend the authors include this application in the review paper to enhance its quality.
Author Response
Author’s Reply to Reviewer No 1 Comments and Suggestions for Authors
In this paper, the authors attempt to review the available studies related to 3D-printed polymeric molds and evaluate their properties and applications. Considering the increasing usage of 3D-printed molds, this subject is interesting and I’m sure it will be insightful for the journal’s readers. However, there are some points that the authors need to address before suggesting it for publication.
Authors’ reply: Dear reviewer, thank you for your effort and kindness in accepting to review our manuscript, as well as for your time and pertinent observations. We have implemented the suggested modification and additions to our manuscript, complementing the references list with a total number of 53 new references (marked with red in the list of references), addressing each of the mentioned points, as follows:
1- In Figure 1, the authors discuss the advantages and disadvantages of additive manufacturing (AM). They mention fast production time and cost-effectiveness as advantages of AM compared to conventional methods. However, in general, this is not true, especially when compared to other methods such as injection molding, as AM often takes more time, especially in moderate and large-scale production. The same can be claimed for cost. Additionally, one of the main disadvantages of AM is that 3D-printed parts may lack sufficient mechanical properties. Please consider revising this figure.
Authors’ reply: Thank you for your pertinent observations related to the figure. Indeed, it required more clear and correct comparison of the advantages versus disadvantages, therefore we have corrected the figure and replaced it with the updated version and added additional text sections, commenting all the features that can act as advantages or disadvantages depending on the technologies and products requirements.
2- To enhance the quality of the information presented in Table 1, please consider adding the wire arc and DED (Directed Energy Deposition) methods as important techniques for metal additive manufacturing (AM).
Authors’ reply: Thank you for drawing our attention to this technique missing from the table. The DED class of AM was added to Table 1, together with its details (materials used, principle, techniques, advantages, disadvantages) and 2 new bibliographic references, and some details were added/corrected in the PBF class.
3- Some of the abbreviations are defined multiple times. Generally, they need to be defined only at the first occurrence. For example, FDM is defined in Table 1 and also in line 84. FFF is defined in Table 1, and also in lines 85, 86, and 90. Please review all instances of these abbreviations in the paper and correct this issue.
Authors’ reply: Thank you for drawing our attention to this aspect. We have reviewed all the mentioned instances as well as similar others identified and corrected them.
4- It would be beneficial if the authors also discussed the main requirements of materials used for 3D printing molds. Additionally, they should consider discussing the considerations that need to be satisfied in the design of 3D-printed polymeric molds.
Authors’ reply: Thank you for your valuable comment. We have added in the Section 3. Technologies that use molds, a text section dedicated to the specific requirements of the materials to be used as solution for mold manufacturing via 3D printing and a text section describing the factors to be considered in terms of design to obtain the desired product quality via the desired process efficiency.
5- One of the main applications of molds is in composite fabrication. Despite mentioning applications for 3D printed molds, the authors didn’t consider this significant application. I suggest including the application of 3D-printed polymeric molds in composite fabrication. For reference, ULTEM 9085 is commonly used for composite tooling. Additionally, Stratasys is one of the pioneers in the application of 3D-printed molds for composite manufacturing.
Authors’ reply: Thank you for your valuable comment. Indeed, we have somehow not given enough consideration to the use of molds for composite fabrication. Some references to the fabrication of 3D printed molds in the production of composites via vacuum bagging that were mentioned in Sub-Section 4.3. 3D printing of molds for thermoforming and vacuum forming, but indeed the subject deserved a dedicated section. Therefore, we have renamed Sub-Section “4.3. 3D printing of molds for thermoforming” and added a new sub-section “4.4. 3D printing of molds for composite fabrication”, dedicated to the composite application, adding new and proper references presenting this topic approach as well as moving the references related to this subject from previous sub-section from the previous version of the manuscript.
6- In recent years, some companies have utilized 3D printing to produce sacrificial polymeric molds or smart molds for fabricating structures with complex shapes. I recommend the authors include this application in the review paper to enhance its quality.
Authors’ reply: Thank you for your valuable comment! It is indeed a field that intensively uses 3D printing of molds. We have added a new sub-section “4.7. 3D printing of sacrificial molds” in which we made a summary of the most remarkable research in the direction, giving references to a significant number of research works (around 30 most remarkable references were selected from the multitude of available studies).

Reviewer 2 Report
Comments and Suggestions for Authors
The manuscript entitled "The Use of Additive Manufacturing Techniques in the Development of Polymeric Molds: A Review" offers a significant contribution to the body of knowledge in additive manufacturing technologies. The authors have diligently curated an extensive collection of literature and presented it in an accessible and informative manner. While the manuscript is commendable, I have outlined a few minor comments below for consideration:
1. Figure 1: The advantages listed, particularly "quick production time" and "minimal waste," could benefit from further clarification. It's important to note that the production time in additive manufacturing (AM) varies depending on factors such as technology type and complexity of the design. This sentence isn’t too precise, compared to technology, e.g. injection moulding, the production time of one element in AM is much longer. Additionally, while AM generally produces less waste compared to traditional manufacturing methods, it's crucial to acknowledge that certain additive techniques may still generate significant waste, especially in certain applications. A brief discussion on the advantages and disadvantages, with specific references to areas where clarity is lacking, would enhance the comprehensiveness of this section.
2. It would be beneficial to provide a brief description of Fused Deposition Modeling (FDM) or Fused Filament Fabrication (FFF) technology, similar to the detailed explanation provided for mold technology. This would aid readers unfamiliar with the basics of the process in understanding its application in mold development. Similar treatment could be extended to other additive manufacturing technologies, such as Stereolithography (SLA).
3. Expanding the comparison between traditional manufacturing methods and additive manufacturing techniques could enrich the manuscript. Referring to existing literature that offers detailed comparisons would provide readers with a more comprehensive understanding of the advantages and limitations of each approach e.g. Sztorch B., BrzÄ…kalski D., JaÅ‚brzykowski M., Przekop R.E., Processing Technologies for Crisis Response on the Example of COVID-19 Pandemic—Injection Molding and FFF Case Study, Processes 2021, 9, 791.
4. Figure 3 and Figure 5: It is recommended to integrate Figures 3 and 5 within the respective sections of the text to enhance the flow of information and aid in better comprehension.
In summary, the manuscript presents valuable insights into the utilization of additive manufacturing techniques in polymeric mold development. Addressing the aforementioned suggestions would further enhance the clarity and comprehensiveness of the review, making it a strong candidate for publication in the field of additive technologies.
Author Response
Author’s Reply to Reviewer No 2 Comments and Suggestions for Authors
The manuscript entitled "The Use of Additive Manufacturing Techniques in the Development of Polymeric Molds: A Review" offers a significant contribution to the body of knowledge in additive manufacturing technologies. The authors have diligently curated an extensive collection of literature and presented it in an accessible and informative manner. While the manuscript is commendable, I have outlined a few minor comments below for consideration:
Authors’ reply: Dear reviewer, thank you for your effort and kindness in accepting to review our manuscript, as well as for your time and pertinent observations. We have implemented the suggested modification and additions to our manuscript, complementing the references list with a total number of 53 new references (marked with red in the list of references), addressing each of the mentioned points, as follows:
- Figure 1: The advantages listed, particularly "quick production time" and "minimal waste," could benefit from further clarification. It's important to note that the production time in additive manufacturing (AM) varies depending on factors such as technology type and complexity of the design. This sentence isn’t too precise, compared to technology, e.g. injection moulding, the production time of one element in AM is much longer. Additionally, while AM generally produces less waste compared to traditional manufacturing methods, it's crucial to acknowledge that certain additive techniques may still generate significant waste, especially in certain applications. A brief discussion on the advantages and disadvantages, with specific references to areas where clarity is lacking, would enhance the comprehensiveness of this section.
Authors’ reply: Thank you for your pertinent observations and clarification required by the information presented in Figure 1. Indeed, the information was presented in a too general way and seemed to be misleading. We have added some clarifications to the information in the figure, replaced it with a corrected version and also added additional text sections, discussing all the features that can act as advantages or disadvantages depending on the technologies and products requirements.
- It would be beneficial to provide a brief description of Fused Deposition Modeling (FDM) or Fused Filament Fabrication (FFF) technology, similar to the detailed explanation provided for mold technology. This would aid readers unfamiliar with the basics of the process in understanding its application in mold development. Similar treatment could be extended to other additive manufacturing technologies, such as Stereolithography (SLA).
Authors’ reply: Thank you for your valuable comment. We have added additional explanation of the three main 3D printing technologies that use polymers (FDM, SLA and SLS), dedicating a sub-section to each of them, presenting detailed explanation (i.e. the working principle scheme, applications, materials used etc.).
- Expanding the comparison between traditional manufacturing methods and additive manufacturing techniques could enrich the manuscript. Referring to existing literature that offers detailed comparisons would provide readers with a more comprehensive understanding of the advantages and limitations of each approach e.g. Sztorch B., BrzÄ…kalski D., JaÅ‚brzykowski M., Przekop R.E., Processing Technologies for Crisis Response on the Example of COVID-19 Pandemic—Injection Molding and FFF Case Study, Processes 2021, 9, 791.
Authors’ reply: Thank you for your comment and also for pointing out the mentioned paper. We have extended the comparison between the two techniques class, including the paper you suggested, as it was of great interest for the topic our manuscript, together with some other references included.
- Figure 3 and Figure 5: It is recommended to integrate Figures 3 and 5 within the respective sections of the text to enhance the flow of information and aid in better comprehension.
Authors’ reply: Thank you very much for your relevant and pertinent observations related to the two figures. Considering all the information added in this revised form, the integration of the details contained by Figure 3 and Figure 5 into the text section was found to be even more appropriate. Therefore, the figures were taken out of the manuscript and the information was embedded into the text, with proper connections to the statements in their section. We think the text is now more comprehensive in terms of the information flow presentation.
In summary, the manuscript presents valuable insights into the utilization of additive manufacturing techniques in polymeric mold development. Addressing the aforementioned suggestions would further enhance the clarity and comprehensiveness of the review, making it a strong candidate for publication in the field of additive technologies.

Round 2
Reviewer 1 Report
Comments and Suggestions for Authors
The revised version is acceptable for publication.